# NEK1-mediated retromer trafficking promotes blood–brain barrier integrity by regulating glucose metabolism and RIPK1 activation

Huibing Wang[1], Weiwei Qi[2], Chengyu Zou[1], Zhangdan Xie[2], Mengmeng Zhang[2], Masanori Gomi Naito[1], Lauren Mifflin [1], Zhen Liu[2], Ayaz Najafov [1], Heling Pan[2], Bing Shan[2], Ying Li [2], Zheng-Jiang Zhu [2] & Junying Yuan [1,2 ✉]

Loss-of-function mutations in *NEK1* gene, which encodes a serine/threonine kinase, are involved in human developmental disorders and ALS. Here we show that NEK1 regulates retromer-mediated endosomal trafficking by phosphorylating VPS26B. NEK1 deficiency disrupts endosomal trafficking of plasma membrane proteins and cerebral proteome homeostasis to promote mitochondrial and lysosomal dysfunction and aggregation of α-synuclein. The metabolic and proteomic defects of NEK1 deficiency disrupts the integrity of blood–brain barrier (BBB) by promoting lysosomal degradation of A20, a key modulator of RIPK1, thus sensitizing cerebrovascular endothelial cells to RIPK1-dependent apoptosis and necroptosis. Genetic inactivation of RIPK1 or metabolic rescue with ketogenic diet can prevent postnatal lethality and BBB damage in NEK1 deficient mice. Inhibition of RIPK1 reduces neuroinflammation and aggregation of α-synuclein in the brains of NEK1 deficient mice. Our study identifies a molecular mechanism by which retromer trafficking and metabolism regulates cerebrovascular integrity, cerebral proteome homeostasis and RIPK1-mediated neuroinflammation.

[1] Department of Cell Biology, Harvard Medical School, Boston, MA, USA. [2] Interdisciplinary Research Center on Biology and Chemistry, Shanghai Institute of Organic Chemistry, Chinese Academy of Sciences, Shanghai, China. ✉email: junying_yuan@sioc.ac.cn

The NIMA-Related Kinase 1 (*NEK1*) gene encodes a serine/threonine kinase with homology to NIMA (never in mitosis gene-A)[1]. Homozygous LoF (loss-of-function) *NEK1* mutations lead to autosomal-recessive lethal osteochondrodysplasia, a severe early developmental disorder known as short rib polydactyly syndrome type II (SRPS) in humans[2]. In addition, *NEK1* reduction of function variants such as p.Arg261His has been identified as a risk factor associated with approximately 3% of ALS cases[3]. NEK1 has been implicated in mediating diverse cellular functions, including cell cycle, cilia formation, DNA-damage response, microtubule stability, and neural development[4]. It remains unclear how NEK1 may mediate these diverse functions at cellular levels; nor is it known why NEK1 deficiency in animals and humans can lead to early developmental disorders. *Nek1*[Kat2J] mutant mice harbor a spontaneous mutation resulting from the insertion of a guanosine residue in the NEK1 kinase coding region which produces a frame shift that results in a premature termination codon (PTC) and leads to a null phenotype[5]. *Nek1*[Kat2J] mutant mice exhibit premature lethality and pleiotropic defects such as facial dysmorphism, polycystic kidney disease, craniofacial anomalies, and growth retardation[6] and thus, provide a model for investigating the null phenotype of *Nek1*. Knockdown of *Nek1* has been found in a focused siRNA screen to sensitize to RIPK1-dependent apoptosis (RDA) by promoting the activation of RIPK1[7]. However, the mechanism by which NEK1 regulates RIPK1 is unknown.

RIPK1 is a serine/threonine kinase which functions as a key regulator of cellular responses in cells stimulated by TNF[8, 9]. RIPK1 contains an N-terminal kinase domain, an intermediate domain, and a C-terminal death domain (DD). Activation of RIPK1 downstream of TNFR1 signaling plays an important role in promoting programmed cell death via either necroptotic or apoptotic pathway. In TNF stimulated cells, RIPK1 is rapidly recruited into complex I in association with the intracellular DD of TNFR1 where RIPK1 is extensively modulated by ubiquitination and phosphorylation which collectively decides whether RIPK1 is to be activated. RIPK1, once activated, interacts with either FADD and caspase-8 to form complex IIa and mediate RIPK1-dependent-apoptosis (RDA) or with RIPK3 to form complex IIb and mediate necroptosis. The activation of RIPK1 is extensively modulated by ubiquitination[9]. A20 is a ubiquitin editing enzyme that is known to modulate Lys63 ubiquitination and activation of RIPK1 to inhibit apoptosis and necroptosis[10–12].

The retromer complexes are highly conserved eukaryotic protein machinery that binds to the cytosolic side of endosomes and mediate retrograde transport from endosomal compartments to the *trans*-Golgi network (TGN) and the plasma membrane[13]. The mammalian retromer complex consists of a heterotrimeric core, VPS26, VPS29, and VPS35, in association with a dimer from the sorting nexin (SNX) family[14]. Mammalian cells express two different isoforms of VPS26: VPS26A and VPS26B. Although VPS26B shares marked homology with VPS26A, its C-terminus is distinct and the functional role of VPS26B is unexplored[15]. Sorting nexin 2 (SNX2) is a mammalian ortholog of yeast retromer component VPS5p and involved in intracellular degradative and recycling functions[16]. The sorting nexin 27 (SNX27)-retromer complex is a major regulator of endosome-to-plasma membrane recycling of transmembrane cargos[17]. SNX27-retromer maintains the cell surface levels of glucose transporter GLUT1 by preventing its lysosomal degradation[18]. Reduced retromer function has been implicated in neurodegenerative diseases such as late-onset Alzheimer's disease, as has decreased glucose metabolism in the brains of patients[19]. Genome-wide association studies (GWAS) have identified many variants in the core components of retromer complexes as risk factors in promoting age-related neurodegenerative diseases, including Alzheimer's disease (AD) and Parkinson's disease (PD)[20]. However, it remains unclear how regulation of retromer is involved in neurodegenerative diseases.

Here we investigated the null phenotype of *Nek1* and the mechanism by which NEK1 deficiency promotes early lethality using *Nek1*[Kat2J] mutant mice. We show that NEK1 regulates retromer-mediated trafficking by directly phosphorylating VPS26B. NEK1 deficiency impairs retromer-mediated trafficking of plasma membrane proteins such as GLUT1 and promotes metabolic defects. Dysregulated metabolism in *Nek1*[Kat2J] cells sensitizes RIPK1 activation by down-regulating A20 which can be inhibited by genetic inactivation of RIPK1 and by metabolic regulation. We demonstrate that the postnatal lethality of *Nek1*-[Kat2J] mutant mice is mediated by BBB damage involving both RDA and necroptosis of cerebrovascular endothelial cells, which can be rescued by either inhibiting RIPK1 activation or rescuing metabolic defects with a ketogenic diet.

## Results

### Rescue of BBB damage in *Nek1*[Kat2J] mice by RIPK1 inhibition.

We first analyzed the expression pattern of NEK1 in the central nervous system (CNS) using immunostaining with anti-NEK1 antibody. In the mouse spinal cord, NEK1 was found in the NeuN+ neurons in the ventral and lateral horns (Supplementary Fig. 1a). A large fraction of ChAT+ motor neurons in the ventral horn of wild type (WT) spinal cords were NEK1+ (Supplementary Fig. 1b). In addition, NEK1 was detected in CD31+ endothelial cells and IBA1+ microglia in WT mouse brains (Supplementary Fig. 1c-d). The immunostaining signal of NEK1 was absent in *Nek1*[Kat2J/Kat2J] mouse brains, consistent with the null phenotype of this mutant line[5].

As reported, approximately half of *Nek1*[Kat2J/Kat2J] newborn pups die before weaning age[6]: the percentage of surviving *Nek1*[Kat2J/Kat2J] pups at 2 weeks of age from intercrosses of *Nek1*[Kat2J/+] mice was only 13.7%. The remaining pups also died early: 6 of the 7 surviving *Nek1*[Kat2J/Kat2J] pups died between 2 to 4 weeks of age and the last one died before 6 weeks of age (Fig. 1a, b). To examine the role of RIPK1 in *Nek1*[Kat2J/Kat2J] mice, we generated *Nek1*[Kat2J/Kat2J];*Ripk1*[D138N/D138N] mice by crossing *Nek1*[Kat2J/+] mice with RIPK1 kinase-dead knock-in *Ripk1*[D138N/D138N] mice[21]. The survival of *Nek1*[Kat2J/Kat2J];*Ripk1*[D138N/D138N] newborns was increased compared to that of *Nek1*[Kat2J/Kat2J] newborns: *Nek1*[Kat2J/Kat2J];*Ripk1*[D138N/D138N] newborns account for 20% of the total littermates that survived past 2 weeks (25% is expected based on Mendelian ratios), compared to 13.7% of littermate *Nek1*[Kat2J/Kat2J] newborns. Among the 15 surviving *Nek1*[Kat2J/Kat2J];*Ripk1*[D138N/D138N] mice, 8 died before 6 weeks, the remaining 7 *Nek1*[Kat2J/Kat2J];*Ripk1*[D138N/D138N] mice survived more than 5 months until they had to be euthanized due to malocclusion caused by craniofacial abnormality between the ages of 5 to 11 months (Fig. 1a, b). Thus, inhibition of RIPK1 kinase extended the postnatal survival of *Nek1*[Kat2J/Kat2J] mice. Inhibition of RIPK1, however, did not affect the polycystic kidney development, dwarfism, or facial deformity in this mutant line (Supplementary Fig. 1e-g). Inhibition of RIPK1 did not rescue the reduced body weight in *Nek1*[Kat2J/Kat2J];*Ripk1*[D138N/D138N] mice at 2 weeks of age (Supplementary Fig. 1h). While inhibition of RIPK1 rescued the lethality of *Nek1*[Kat2J/Kat2J] mice to allow their survival, the body weight of *Nek1*[Kat2J/Kat2J];*Ripk1*[D138N/D138N] mice at 5 months of age was still lower than that of WT or *Ripk1*[D138N/D138N] mice (Supplementary Fig. 1i). Thus, the activation of RIPK1 is involved in mediating postnatal lethality but not developmental defects in *Nek1* deficient mice.

We next explored the mechanism by which NEK1 deficiency promotes the postnatal lethality. Activation of RIPK1 can mediate both RIPK1-dependent apoptosis (RDA) and necroptosis[8]. We

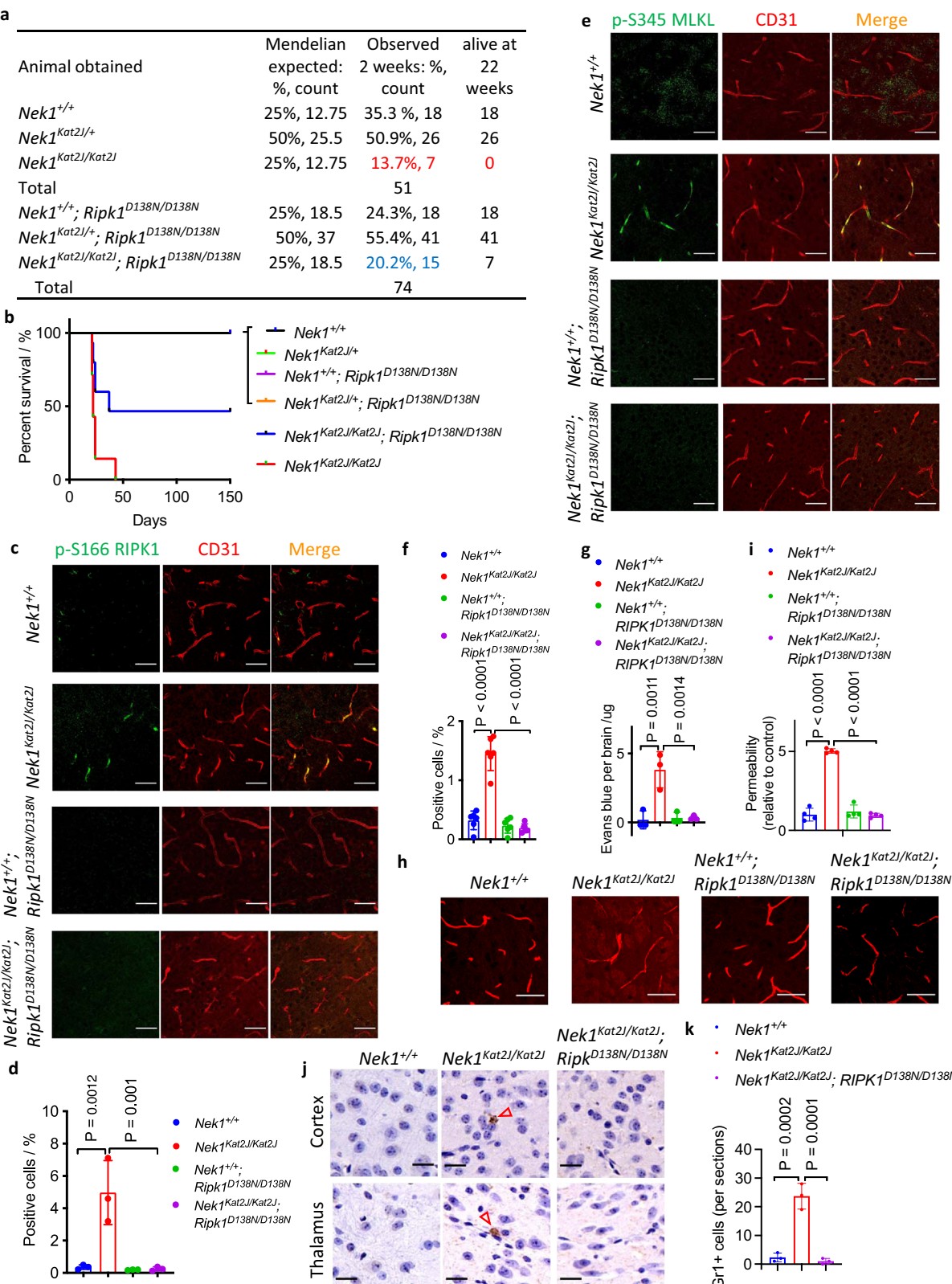

analyzed the biomarkers of RIPK1 activation (p-S166), necroptosis (p-S345 MLKL) and apoptosis (cleaved caspase-3) by immunostaining in the brains of WT and Nek1$^{Kat2J/Kat2J}$ mice at 3 weeks of age. We found that a substantial amount of p-S166 RIPK1, p-S345 MLKL as well as cleaved caspase-3 signals in the brains of Nek1$^{Kat2J/Kat2J}$ mice, which colocalized with CD31$^{+}$ cerebrovascular endothelial cells. Inhibition of RIPK1 blocked the immunostaining signals of

p-S166 RIPK1, p-S345 MLKL, and cleaved caspase-3 in the brains of Nek1$^{Kat2J/Kat2J}$;Ripk1$^{D138N/D138N}$ mice at 3 weeks of age (Fig. 1c–f and Supplementary Fig. 1j). These data suggest that Nek1 deficiency promotes the activation of RIPK1, necroptosis, and apoptosis in cerebrovascular endothelial cells.

Since the activation biomarkers of apoptosis and necroptosis were predominantly associated with cerebrovascular endothelial

**Fig. 1 Rescue of BBB damage in *Nek1*Kat2J mice by RIPK1 inhibition. a**, **b** The surviving progenies (at 2 weeks and 22 weeks of age) from the following matings: 8 female *Nek1*$^{Kat2J/+}$ mice were mated with 4 male *Nek1*$^{Kat2J/+}$ mice; 8 female *Nek1*$^{Kat2J/+}$; *Ripk1*$^{D138N/D138N}$ mice were mated with 4 male *Nek1*$^{Kat2J/+}$; *Ripk1*$^{D138N/D138N}$ mice. The % of survival was ploted (**b**). **c**–**f**, Cerebral cortical sections from the mice (3 weeks old, female) were immunostained with anti-*p*-S166 RIPK1 (**c**) and anti-p-S345 MLKL (**e**). The fluorescent signals of p-S166 RIPK1 (**d**) and p-S345 MLKL (**f**) were quantified. Mean ± SD. $n = 3$ sections per group for p-S166 RIPK1. $n = 6$ sections per group for p-S345 MLKL. Bar = 50 μm. One-way ANOVA with Dunnett's test. **g** Mice (3 weeks old) were inraperitoneally injected with 2% (w/v) Evans blue solution (70 mg/kg) and followed by PBS perfusion. Extravasated Evans blue levels in total brain lysates were measured by a plate reader. Mean ± SD. $n = 3$ mice per group. One-way ANOVA with Dunnett's test. **h**, **i** Mice (3 weeks old) were intravenously injected with 10kD-tetramethyrhodamine (TMR)-labeled Dextran and then sacrificed. The cryo-sections of cerebral cortex were imaged by confocal microscope. Bar = 20 μm. Fluorescent intensity of diffused dextran around cerebral vessels was quantified using ImageJ. The fluorescent intensity of all groups was nomalized to *Nek1*$^{+/+}$ mice (mean ± SD, $n = 4$ sections per group) and graphed in (**i**). One-way ANOVA with Dunnett's test. **j**, **k** Representative images of Gr1 immunohistostaining (brown) of cerebral cortex and thalamus sections (3 weeks old mice). Bar = 20 μm. Number of Gr1 positive cells per sections were presented as mean± SD. $n = 3$ sections per group. One-way ANOVA with Dunnett's test.

cells in *Nek1*$^{Kat2J/Kat2J}$ mice, we next investigated the integrity of blood–brain barrier (BBB) in *Nek1*$^{Kat2J/Kat2J}$ mice and the effect of RIPK1 inhibition. We measured the integrity of BBB using Evans blue[22]. Littermate *Nek1*$^{Kat2J/Kat2J}$, *Nek1*$^{Kat2J/Kat2J}$; *Ripk1*$^{D138N/D138N}$, and WT mice (~3 weeks old) were intraperitoneally injected with 2% Evans blue and sacrificed 24 h later. The brains of *Nek1*$^{Kat2J/Kat2J}$ mice showed a substantial amount of extravasated Evans blue, whereas WT littermates, *Nek1*$^{+/+}$; *Ripk1*$^{D138N/D138N}$ mice and *Nek1*$^{Kat2J/Kat2J}$;*Ripk1*$^{D138N/D138N}$ mice showed no extravasated Evans blue (Fig. 1g). The BBB integrity was further evaluated with intravenous injection of rhodamine-labeled dextran (MW: 10 kDa) into the tail vein of littermate *Nek1*$^{Kat2J/Kat2J}$ mice, *Nek1*$^{Kat2J/Kat2J}$;*Ripk1*$^{D138N/D138N}$ mice, and WT mice[23]. While dextran was exclusively confined to cerebromicrovessels in WT mice, diffused dextran into the cerebral parenchyma was observed in *Nek1*$^{Kat2J/Kat2J}$ mice (Fig. 1h, i). In both *Nek1*$^{+/+}$;*Ripk1*$^{D138N/D138N}$ mice and *Nek1*$^{Kat2J/Kat2J}$;*Ripk1*$^{D138N/D138N}$ mice, injected dextran was also confined to cerebromicrovessels. BBB damage can lead to neutrophil infiltration into brain parenchyma[22]. Infiltrated Gr1$^+$ neutrophils were found in the brain parenchyma of *Nek1*$^{Kat2J/Kat2J}$ mice, but not WT mice, and were reduced in *Nek1*$^{Kat2J/Kat2J}$;*Ripk1*$^{D138N/D138N}$ mice (Fig. 1j, k). Thus, *Nek1* deficiency leads to early impairment in the BBB integrity and neutrophil infiltration, which can be inhibited by blocking RIPK1 kinase.

We next investigated the vulnerability of *Nek1*-deficient cells to apoptosis and necroptosis. Treatment of *Nek1*$^{Kat2J/Kat2J}$ MEFs, but not *Nek1*$^{+/+}$ MEFs, with TNF induced cell death, which was prevented by RIPK1 inhibitor Nec-1s (R-7-Cl-O-Nec-1) (Supplementary Fig. 2a). *Nek1*$^{Kat2J/Kat2J}$ MEFs were also sensitized to RDA induced by a combination of TNF and TAK1 inhibitor, 5z7 (5z-7-Oxozeaenol) and necroptosis induced by TNF/5z7/zVAD (Supplementary Fig. 2b, c). Knockdown of *Nek1* using shRNA also sensitized RGC5 cells and L929 cells to RDA and necroptosis, which was protected by RIPK1 inhibition with Nec-1s (Supplementary Fig. 2d-f). Thus, *Nek1* deficiency sensitizes MEFs to both RDA and necroptosis.

**Downregulation of A20 in *Nek1*$^{Kat2J}$ cells.** We next characterized the mechanism by which *Nek1* deficiency sensitizes to RIPK1-dependent cell death by analyzing the components of the TNFR1 signaling complex (TNF-RSC or complex I) which forms rapidly in TNF-stimulated cells and is crucial in determining cell fate[24]. Interestingly, we found that the levels of A20 recruited to complex I were much lower in *Nek1*$^{Kat2J/Kat2J}$ cells relative to that of WT (Fig. 2a). The levels of activation (p-S166) and Lys63 ubiquitination of RIPK1 in complex I were correspondingly higher in *Nek1*$^{Kat2J/Kat2J}$ MEFs than that of WT (Fig. 2a, b). Furthermore, we found that the levels of total A20 protein were much lower in *Nek1*$^{Kat2J/Kat2J}$ cells relative to WT, but A20 mRNA levels were

not affected by *Nek1* deficiency (Fig. 2a; Supplementary Fig. 2g). The reduced A20 levels could be rescued by expression of WT NEK1, but not kinase dead D146N NEK1 mutant in *Nek1*$^{Kat2J/Kat2J}$ MEFs, indicating that A20 protein levels were impacted by NEK1 kinase activity (Fig. 2c, d). Consistent with A20 deficiency[25], the phosphorylation and degradation of IκBα in *Nek1*$^{Kat2J/Kat2J}$ MEFs stimulated by TNF was increased compared to that of WT (Supplementary Fig. 2h). Treatment with lysosome inhibitors E64d and chloroquine (CQ), but not proteasomal inhibitor MG132, rescued the reduced levels of A20 protein in *Nek1*$^{Kat2J/Kat2J}$ MEFs (Fig. 2e, f). These results suggest that lysosomal degradation of A20 is enhanced in *Nek1*$^{Kat2J/Kat2J}$ MEFs. Since A20 deficiency sensitizes to the activation of RIPK1[26], these data suggest that enhanced lysosomal degradation of A20 protein may sensitize to RDA and necroptosis in NEK1 deficient cells.

Treatment with TNF/5z7 to induce RDA accelerated RIPK1 activation (p-S166) and caspase-3 cleavage in *Nek1*$^{Kat2J/Kat2J}$ MEFs to a level much higher than that of WT MEFs, which was effectively blocked by Nec-1s (Fig. 2g). Induction of necroptosis by the combination of TNF/SM164/zVAD also induced higher levels of p-S166 RIPK1, p-T231/S232 RIPK3, and p-S345 MLKL in *Nek1*$^{Kat2J/Kat2J}$ MEFs compared to that of WT, which was blocked by Nec-1s (Fig. 2h). We compared the levels of A20 protein in the cerebrovascular endothelial cells and found a reduction of A20 protein in *Nek1*$^{Kat2J/Kat2J}$;*Ripk1*$^{D138N/D138N}$ compared to *Nek1*$^{+/+}$;*Ripk1*$^{D138N/D138N}$ mice (Fig. 2i). Since A20 deficiency in mouse models phenocopies human inflammatory diseases in a RIPK1-dependent manner[26], these results suggest that reduced levels of A20 may promote the activation of RIPK1 and sensitize to BBB damage in *Nek1*$^{Kat2J/Kat2J}$ mice, which may directly contribute to the peri-natal lethality.

***Nek1* deficiency disrupts proteome homeostasis in the CNS.** Although the early lethality of *Nek1*$^{Kat2J/ Kat2J}$ mice precluded us from a full characterization of *Nek1* deficiency on neurological functions, we looked for cerebral histological pathology in the post-weaning mutant mice. Disruption of proteostasis and accumulation of misfolded proteins in the CNS is a hallmark of neurodegenerative diseases including ALS[27, 28]. We stained brain slices from *Nek1*$^{Kat2J/ Kat2J}$, *Nek1*$^{Kat2J/Kat2J}$;*Ripk1*$^{D138N/D138N}$, and WT littermates with thioflavin S, which marks misfolded protein aggregates (Fig. 3a, b; Supplementary Fig. 3a). Interestingly, Thioflavin S$^+$ (ThS$^+$) aggregates were found in the cortical neurons of *Nek1*$^{Kat2J/Kat2J}$ mice. Such ThS$^+$ aggregates were also found in the cortical neurons of *Nek1*$^{Kat2J/Kat2J}$;*Ripk1*$^{D138N/D138N}$ mice; however, they were smaller and less numerous compared to those in *Nek1*$^{Kat2J/Kat2J}$ mice (Fig. 3a, b). Pathological aggregation of α-synuclein is a pathological hallmark of Parkinson's Disease (PD), multiple system atrophy and dementia with Lewy bodies[29]. Strikingly, we found that the levels of α-synuclein were elevated in the hippocampal and cortical neurons of *Nek1*$^{Kat2J/Kat2J}$ mice

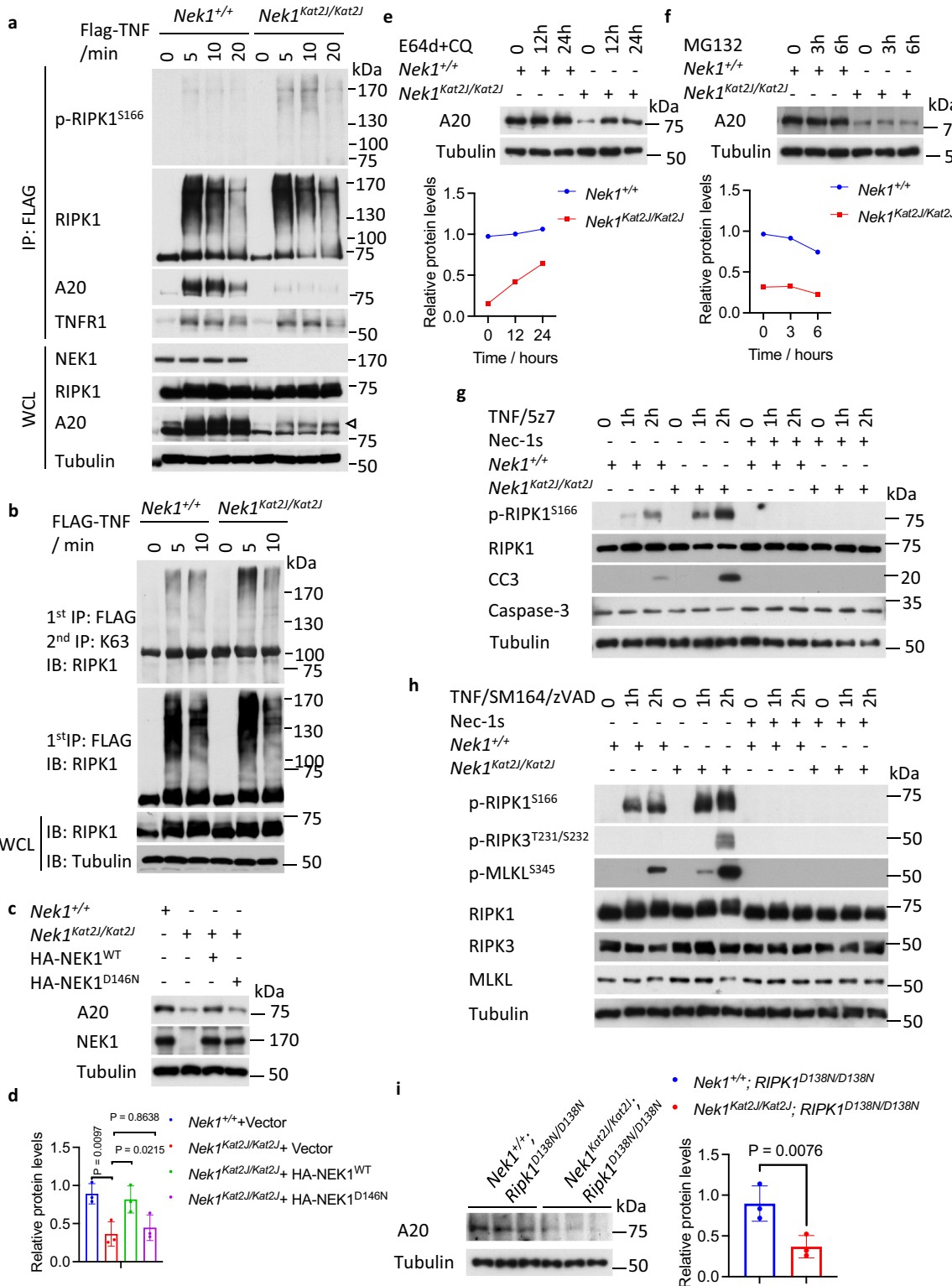

compared to that of WT littermates (Fig. 3c–e; Supplementary Fig. 3b). The protein levels of α-synuclein in the neurons of *Nek1*[Kat2J/Kat2J];*Ripk1*[D138N/D138N] mouse brains were reduced compared to that of *Nek1*[Kat2J/Kat2J] mice (Fig. 3c–e).

We characterized the mRNA levels of proinflammatory cytokines and chemokines in primary microglia isolated from newborn pups. Increased mRNA levels of *Tnf*, *Il1α*, *Il1β*, *Il6*, *Ccl3*, and *Ccl4* were observed in *Nek1*[Kat2J/Kat2J] primary microglia

compared to primary microglia WT littermates (Supplementary Fig. 3c). We next investigated the mRNA levels of proinflammatory cytokines in both spinal cords and brains of *Nek1*[Kat2J/Kat2J] mice. Compared to their WT littermates, the levated mRNA levels of *Cxcl1*, *Ccl2*, *Ccl5*, and *Il1β* were elevated in the spinal cords of *Nek1*[Kat2J/Kat2J] mice; furthermore, the mRNA levels of *Tnf*, *Il1α*, *Il1β*, *Il6*, *Cxcl1*, and *Ccl2* were also increased in *Nek1*[Kat2J/Kat2J] brains (Fig. 3f, g). The increased levels of mRNA

**Fig. 2 Down-regulation of A20 in Nek1$^{Kat2J}$ cells. a** WT and Nek1$^{Kat2J/Kat2J}$ MEFs were treated with FLAG-TNF 50 ng/mL for indicated periods of time. Whole cell lysates and complex I of the treated MEFs were analyzed by western blotting with indicated antibodies. Uncropped blots in the Source Data file. **b** WT and Nek1$^{Kat2J/Kat2J}$ MEFs were treated with FLAG-TNF 50 ng/mL for indicated periods of time and then lysed with NP40 buffer. The cell lysates were immunoprecipitated with anti-FLAG antibody conjugated agarose (1$^{st}$ IP). The immunoprecipitated proteins were eluted and immunoprecipitated with anti-K63 ubiquitin antibody[76] (2$^{nd}$ IP). Uncropped blots in the Source Data file. **c, d** WT and Nek1$^{Kat2J/Kat2J}$ MEFs were transfected with expression vectors of HA-NEK1, HA-NEK1$^{D146N}$ or vector for 72 h and then the cells were lysed and analyzed by western blotting using indicated abs (**c**) and quantified with ImageJ (**d**). Uncropped blots in the Source Data file. n=3 replicates for each group. One-way ANOVA, Dunnett's test. **e** Nek1$^{+/+}$ and Nek1$^{Kat2J/Kat2J}$ MEFs were treated with 25 μM chloroquine (CQ) + 5 μg/mL E64d for 12 and 24 h. Uncropped blots in the Source Data file. **f** Nek1$^{+/+}$ and Nek1$^{Kat2J/Kat2J}$ MEFs were treated with 10 μM MG132 for 3 and 6 h. The levels of A20 were analyzed by western blotting with indicated antibodies and quantified using ImageJ (bottom). Uncropped blots in the Source Data file. **g** Nek1$^{+/+}$ and Nek1$^{Kat2J/Kat2J}$ MEFs were pretreated with 10 μM Nec-1s for 30 min and then treated with 1 ng/mL TNF + 500 nM 5z7 for indicated time. Cell lysates were analyzed by western blotting with indicated antibodies. Uncropped blots in the Source Data file. **h** Nek1$^{+/+}$ and Nek1$^{Kat2J/Kat2J}$ MEFs were pre-treated with 10 μM Nec-1s + 50 nM SM164 for 30 min and then treated with 10 ng/mL TNF + 20 μM zVAD.fmk for indicated periods of time. Cell lysates were analyzed by western blotting with indicated antibodies. Uncropped blots in the Source Data file. **i** Cerebromicrovessels were isolated from the brains of Nek1$^{+/+}$; Ripk1$^{D138N/D138N}$ and Nek1$^{Kat2J/Kat2J}$; Ripk1$^{D138N/D138N}$ mice. The brains of two 10-month-old mice of each genotype were pooled together for the isolation of cerebromicrovessels which were analyzed by western blotting and quantified using ImageJ. n = 3 repeated blots. Data are presented as mean ± SD. Two-tailed Student's t test. Uncropped blots in the Source Data file.

levels of proinflammatory chemokines and cytokines in Nek1-$^{Kat2J/Kat2J}$ mice were suppressed by inhibition of RIPK1 in Nek1$^{Kat2J/Kat2J}$;Ripk1$^{D138N/D138N}$ mice (Fig. 3f, g). Thus, Nek1 deficiency promotes RIPK1-dependent inflammation in the CNS.

The expression of Cst7, a gene that encodes Cystatin F, an endosomal/lysosomal cathepsin inhibitor known as a biomarker for disease-associated microglia (DAMs)[30], was also highly elevated (16-58x) in the brains and spinal cords of Nek1$^{Kat2J/Kat2J}$ mice (Fig. 3f, g). In contrast, the expression of Cst7 in primary microglia isolated from Nek1$^{Kat2J/Kat2J}$ mice was increased by 1.48x (Supplementary Fig. 3c). Cst7 is highly induced in the CNS in microglia from the SOD1$^{G93A}$ ALS mouse model[31], human Alzheimer's disease brains (AD), the APP/PS1 mouse model of AD[32], aging microglia, during demyelination[33], and in a prion disease model[34]. These data suggest that there is an induction of DAM in the CNS of Nek1$^{Kat2J/Kat2J}$ mice.

Consistent with the induction of DAM, the expression of all other known DAM signature genes including ApoE, Trem2 (6.14x), Tyrobp(4.26x), Axl, Ctsl, Cd9, Csf1, Clec7a(77.17x), Lilrb4 (44.81x), Ctsd, Ctsb and Itgax(10.79x), except Timp2, were also highly elevated in the brains of Nek1$^{Kat2J/Kat2J}$ mice, which were suppressed by the inhibition of RIPK1 in Nek1$^{Kat2J/Kat2J}$; Ripk1$^{D138N/D138N}$ mice (Supplementary Fig. 3d). In particular, some of these RIPK1-regulated DAM markers, such as Apoe[35], Trem2[36], Tyrobp[37], and Ctsd[38], are known AD risk factors. In contrast, the markers of homeostatic microglia, such as Tmem119, R2ry12, Cx3cr1, Sall1, and Rhob[30], were not elevated, or minimally induced, such as Gpr34(1.85x), Fcrls(3.61x) and Olfml3(3.04x), in the brains of Nek1$^{Kat2J/Kat2J}$ mice, which were also suppressed by the inhibition of RIPK1 in the brains of Nek1$^{Kat2J/Kat2J}$;Ripk1$^{D138N/D138N}$ mice (Supplementary Fig. 3e). Taken together, these data support that Nek1 deficiency disrupts proteome homeostasis which leads to the formation of α-synuclein aggregates and activates microglial inflammation and DAM in RIPK1-dependent manner.

**NEK1 regulates retromer function by phosphorylating VPS26B.** To investigate the molecular mechanism by which Nek1 regulates proteome homeostasis, we compared the list of NEK1 binding proteins identified by mass spectrometry analysis[39] with hits from a genome-wide screen of regulators of RDA and necroptosis[7, 40]. We found that NEK1 bound to VPS26B and VPS35 (Supplementary Fig. 4a) and furthermore, knockdown of VPS26B and VPS29 sensitized cells to necroptosis (Supplementary Fig. 4b). The trimer of VPS26B, VPS29, and VPS35 binds with sorting nexin (SNX) proteins in a complex known as the

retromer which plays an important role in recycling proteins from endosomes to the trans-Golgi network and plasma membrane[41]. Furthermore, we found that knockdown of VPS26B sensitized WT, but not Nek1$^{Kat2J/Kat2J}$ MEFs, to RDA and necroptosis; while knockdown of VPS29 or VPS26A sensitized both WT and Nek1$^{Kat2J/Kat2J}$ MEFs to RDA and necroptosis, which was inhibited by treatment with RIPK1 inhibitor Nec-1s (Fig. 4a, b; Supplementary Fig. 4c-f). These data suggest that NEK1 targets VPS26B in retromer complex to regulate necroptosis and RDA.

To investigate the global effect of NEK1 deficiency in cells on the retromer functions, we conducted quantitative mass spectrometry analysis of plasma membrane proteins in Nek1$^{+/+}$ and Nek1$^{Kat2J/Kat2J}$ MEFs captured by cell surface selective aminooxy-biotinylation[42]. Our analysis identified statistically significant changes in the membrane levels of 359 proteins among 2688 analyzed plasma membrane proteins between Nek1$^{+/+}$ and Nek1$^{Kat2J/Kat2J}$ MEFs. Among the 359 membrane proteins whose plasma membrane levels were changed in Nek1$^{Kat2J/Kat2J}$ MEFs, 80 (22%) were known retromer components or substrates (Fig. 4c, d)[18]. These data suggest that Nek1 deficiency leads to a systemic disturbance in the retromer-mediated trafficking and plasma membrane localization of the retromer substrates.

Sorting nexin protein SNX27 is known to be involved in the retromer-mediated retrieval of membrane proteins from endosome to plasma membrane[18]. In our mass spectrometry analysis of cell surface proteins, the plasma membrane levels of SNX27 were much higher in Nek1$^{Kat2J/Kat2J}$ MEFs compared to WT MEFs, while the plasma membrane levels of another sorting nexin protein SNX2, a retromer binding protein that regulates the retrieval of proteins from endosome to Golgi apparatus[16], was much lower in Nek1$^{Kat2J/Kat2J}$ compared to WT MEFs (Supplementary Fig. 4g).

We next investigated if NEK1 regulated the phosphorylation of retromer components. Using mass spectrometry analysis, we identified that NEK1 could phosphorylate Ser302 and Ser304 of VPS26B (Supplementary Fig. 4h). We confirmed that phosphorylation of endogenous VPS26B protein was reduced in Nek1$^{Kat2J/Kat2J}$ compared to WT MEFs (Fig. 4e). Using an in vitro kinase assay, we found that VPS26B could be directly phosphorylated by NEK1 in vitro, and this phosphorylation was Ser302/Ser304 dependent as the phosphorylation was blocked by VPS26B-S302A/S304A mutation (Fig. 4f).

We next investigated the functional significance of VPS26B phosphorylation by NEK1. Although the overall structure of VPS26B is highly similar to VPS26A, VPS26B contains an unique conformationally flexible "tail" (a.a. residues 297-336) that

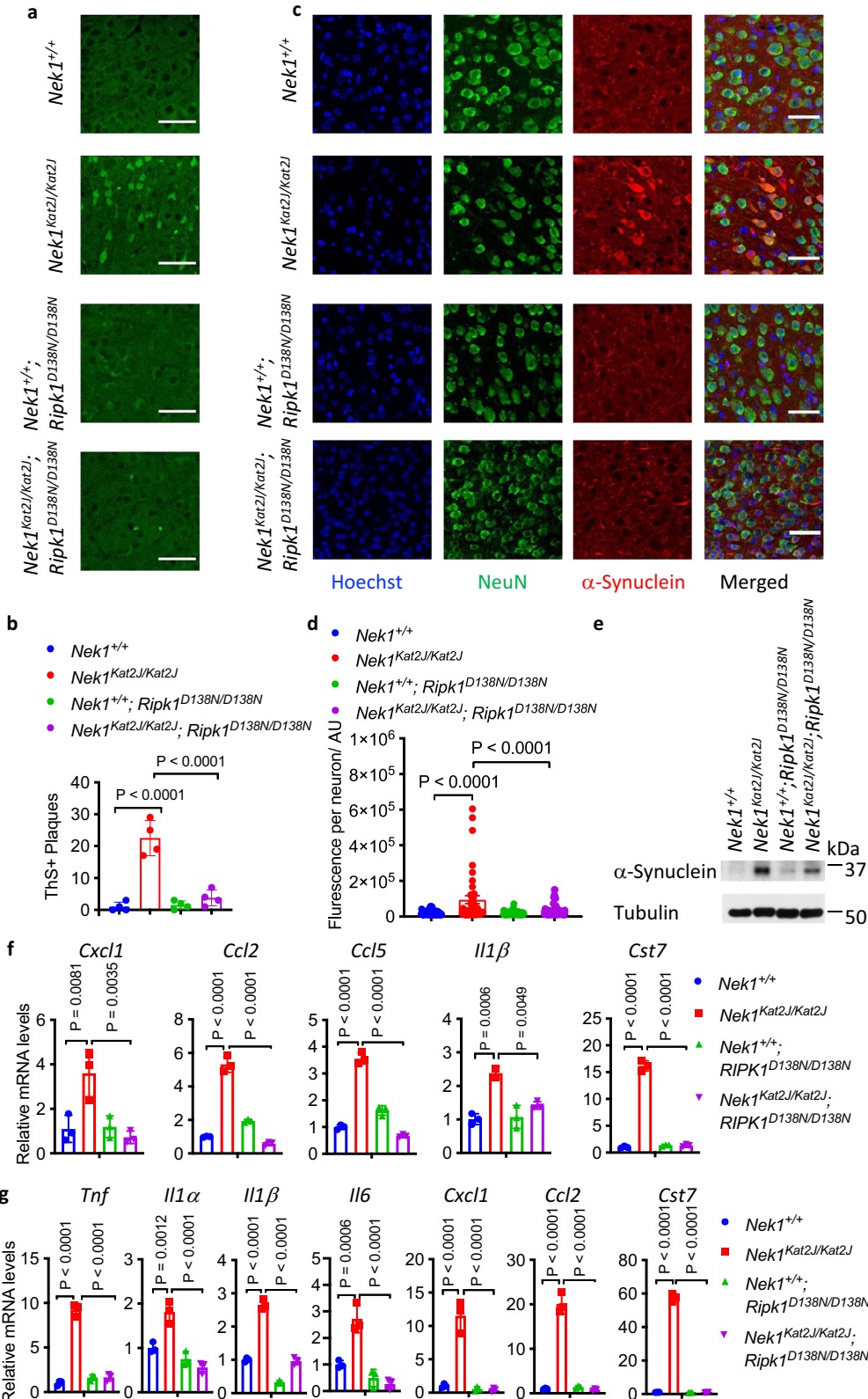

extends from the center of the protein, which has no structural information available[43]. We found that the C-terminus of VPS26B is important for binding between VPS26B and SNX27 as a truncation of VPS26B that lacks the C-terminal 298-336 residues bound poorly with endogenous SNX27 (Supplementary Fig. 4i). The binding of SNX27 with a phosphomimetic VPS26B S302D/S304D double mutant was reduced, while the binding of

SNX27 with a non-phosphomimetic VPS26B S302A/S304A double mutant was increased (Fig. 4g, h); while the binding of SNX2 with VPS26B S302A/S304A was decreased (Fig. 4g, i). In contrast, the binding of SNX27 or SNX2 with VPS26B was not affected by S311A or S311D mutation (Fig. 4g). Addback of VPS26B^WT and VPS26B^S302D/S304D, but not VPS26B^S302A/S304A, rescued VPS26B knockdown cells from the sensitization to cell

**Fig. 3 Nek1 deficiency disrupts proteome homeostasis in the CNS. a, b** Cerebral cortical sections from mice (3 weeks old) with indicated genotypes were stained with Thioflavin S for protein aggregates and quantified (**b**). Mean ± SD. n = 4 sections. One-way ANOVA with Dunnett's test. Bar = 50 μm. **c, d** Cerebral cortical sections of mice (3 weeks old) with indicated genotypes were immunostained with anti-α-synuclein(red) and anti-NeuN(green); and the fluorescent intensity of α-synuclein in each NeuN positive cells was quantified (**d**). Bar = 50 μm. n = 3 sections per group. Data are presented as mean ± SEM. One-way ANOVA with Dunnett's test. **e** Total cortical lysates from Nek1^(+/+) mice, Nek1^(Kat2J/Kat2J) mice, Nek1^(+/+); Ripk1^(D138N/D138N) mice and Nek1^(Kat2J/Kat2J);Ripk1^(D138N/D138N) mice were analyzed by western blotting with indicated antibodies. Uncropped blots in the Source Data file. **f, g** The mRNAs from spinal cords (**f**) or brains (**g**) mice with indicated genotypes (n = 3 per group, 3 weeks old) were extracted and quantified for the levels of indicated cytokines by qPCR. Data are presented as mean ± SD. One-way ANOVA with Dunnett's test.

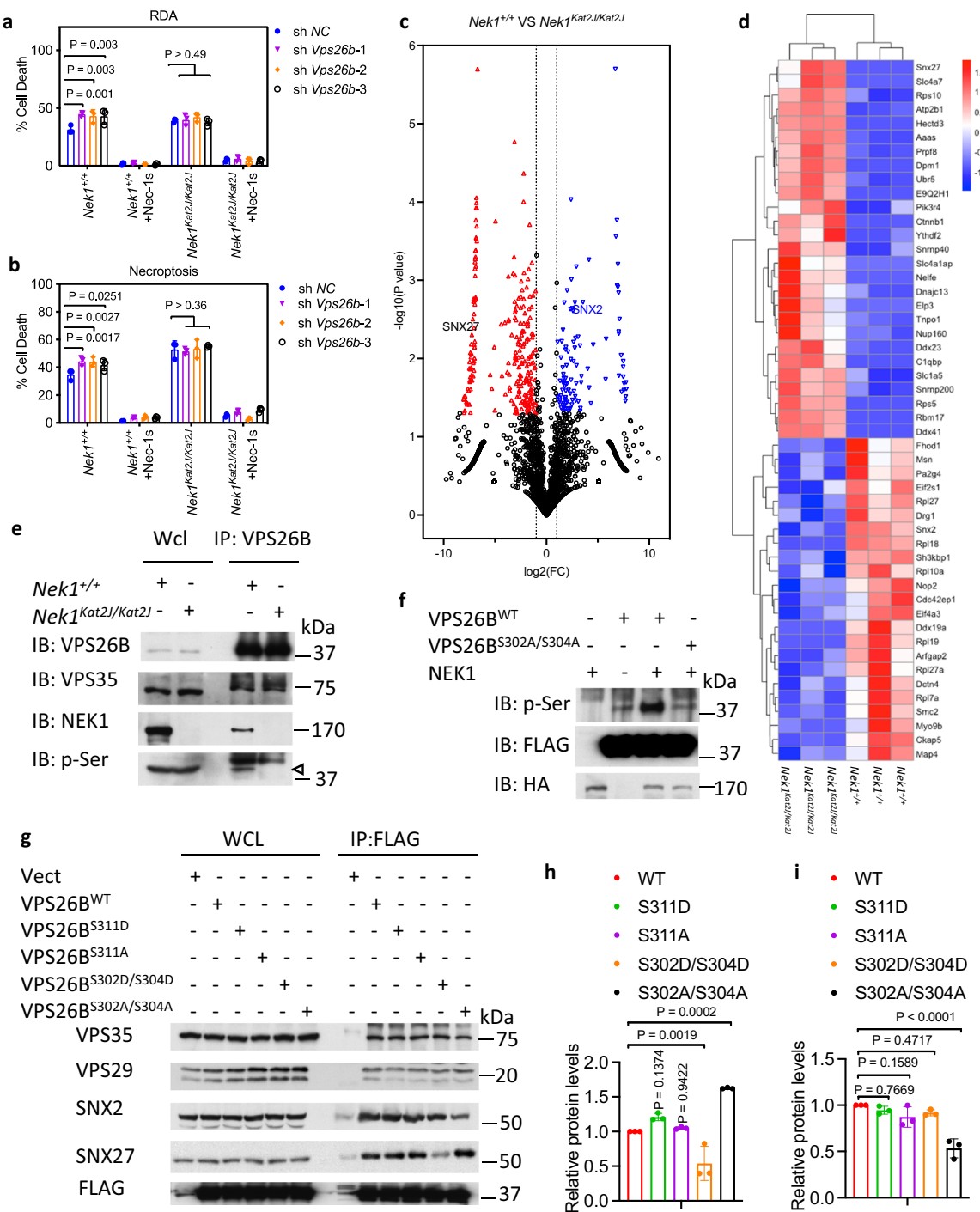

**Fig. 4 NEK1 regulates retromer function by mediating the phosphorylation of VPS26B. a, b** $Nek1^{+/+}$ and $Nek1^{Kat2J/Kat2J}$ MEFs were transfected with shRNA targeting *Vps26b* or scrambled shRNA for 7 days. The cells were then pretreated with 10 μM Nec-1s for 30 min, followed by treatment with 1 ng/mL TNF + TAK1 inhibitor 500 nM 5z 7-oxozeaenol (5z7) (**a**) or 10 ng/mLTNF + 50 nM SM164 + 20 μM zVAD.fmk (**b**) for 5 h. The cell death was measured with ToxiLight. Mean ± SD. $n = 3$ biological repeats. Two-way ANOVA with Dunnett's test. **c** Volcano plot analysis of mass spectrometry data comparing the plasma membrane proteins of $Nek1^{+/+}$ and $Nek1^{Kat2J/Kat2J}$ MEFs enriched by Pierce™ Cell Surface Protein Isolation Kit. $n = 3$ biological independent repeats. Unpaired two-tailed Student's t test. **d** Heatmap analysis of known retromer substrates that showed statistically significant difference in the plasma membrane presence between $Nek1^{+/+}$ and $Nek1^{Kat2J/Kat2J}$ MEFs in the mass spectrometry analysis (**c**). **e** $Nek1^{+/+}$ and $Nek1^{Kat2J/Kat2J}$ MEFs were lysed with NP-40 buffer, and cell lysates were immunoprecipitated with anti-VPS26B antibody. Whole cell lysates and immunoprecipitated proteins were analyzed by western blotting with indicated antibodies. Arrowhead points to the molecular weight expected for VPS26B. Uncropped blots in the Source Data file. **f** NEK1 phosphorylates VPS26B in a Ser302/304 dependent manner. 293 T cells were transfected with expression vectors for HA tagged NEK1 and FLAG tagged VPS26B$^{WT}$, or VPS26B$^{S302A/S304A}$ for 24 h. The cells were then lysed with NP-40 buffer and immunoprecipitated with anti-HA or anti-FLAG antibody conjugated agarose. Purified FLAG-tagged VPS26B$^{WT}$ and VPS26B$^{S302A/S304A}$ were incubated with purified HA-NEK1 (30 °C. 1 h) in kinase reaction buffer. The reaction products were analyzed by western blotting with indicated antibodies. Uncropped blots in the Source Data file. **g–i** 293 T cells were transfected with expression vectors for FLAG-tagged VPS26B$^{WT}$, VPS26B$^{S311D}$, VPS26B$^{S311A}$, VPS26B$^{S302D/S304D}$ or VPS26B$^{S302A/S304A}$ for 24 h and then lysed with NP-40 buffer. The cell lysates were immunoprecipitated with anti-FLAG antibody conjugated agarose. The whole-cell lysates and immunoprecipitated proteins were analyzed by western blotting with indicated antibodies (**g**). Uncropped blots in the Source Data file. SNX27 (**h**) and SNX2 (**i**) binding to VPS26B was normalized to immunoprecipitated FLAG-VPS26B and quantified using ImageJ. $n = 3$ repeated experiments and blots. Data are presented as mean ± SD. One-way ANOVA with Dunnett's test.

death stimuli (Supplementary Fig. 4j, k). Taken together, these data suggest that the phosphorylation of VPS26B by NEK1 on Ser302/Ser304 serve important functions in regulating the trafficking of SNX27-retromer and SNX2-retromer.

We next analyzed the effect of VPS26B$^{S302A/S304A}$ and VPS26B$^{S302D/S304D}$ on the trafficking of SNX27 and SNX2. Consistent with increased plasma membrane presence of SNX27 in $Nek1^{Kat2J/Kat2J}$ MEFs compared to that of WT, the levels of SNX27 in the cell surface membrane were increased in $Nek1^{Kat2J/Kat2J}$ MEFs expressing VPS26B$^{S302A/S304A}$ and reduced in cells expressing VPS26B$^{S302D/S304D}$. In contrast, the levels of SNX2 in the cell surface membrane were decreased in VPS26B knockdown MEFs expressing VPS26B$^{S302A/S304A}$ and increased in VPS26B knockdown MEFs expressing VPS26B$^{S302D/S304D}$ (Supplementary Fig. 4l). The levels of WT VPS26B, VPS26B$^{S302A/S304A}$ and VPS26B$^{S302D/S304D}$ expression were not different. In addition, we further confirmed the increased and decreased plasma membrane levels of SNX27 and SNX2, respectively, in $Nek1^{Kat2J/Kat2J}$ MEFs compared to that of WT by western blotting analysis after cell surface selective aminooxy-biotinylation (Fig. 5a). Taken together, these results demonstrate that Nek1 deficiency blocks the phosphorylation of S302/S304 VPS26B which in turn leads to mislocalization of SNX27- and SNX2-retromers.

**NEK1 deficiency disrupts glucose uptake.** SNX27-retromer is known to mediate the transport of glucose transport 1 (GLUT1) from endosomes to the plasma membrane[18]. We compared the levels of GLUT1 protein in plasma membrane after cell-surface biotinylation in $Nek1^{Kat2J/Kat2J}$ and WT MEFs. We found that the plasma membrane distribution of GLUT1 in $Nek1^{Kat2J/Kat2J}$ MEFs was much lower than WT MEFs (Fig. 5a). The reduced plasma membrane localization of GLUT1 in $Nek1^{Kat2J/Kat2J}$ MEFs was further confirmed by immunostaining (Fig. 5b, c). Reduced plasma membrane presence of GLUT1 was not affected by treatment of Nec-1s (Supplementary Fig. 5a), which is consistent with the model that activation of RIPK1 is downstream of retromer defects in Nek1 deficient cells.

The reduced plasma membrane levels of GLUT1 in $Nek1^{Kat2J/Kat2J}$ MEFs were restored by the expression of WT NEK1 but not NEK1$^{D146N}$, a kinase dead mutant of NEK1, suggesting the importance of NEK1 kinase activity in regulating the trafficking of GLUT1 (Supplementary Fig. 5b). To assess if the phosphorylation of VPS26B at Ser302/Ser304 affects the

localization of GLUT1 on plasma membrane, we expressed VPS26B$^{WT}$, VPS26B$^{S302A/S304A}$ and VPS26B$^{S302D/S304D}$ in HeLa cells. The expression of VPS26B$^{WT}$ and VPS26B$^{S302D/S304D}$ promoted the localization of GLUT1 to the plasma membrane (Fig. 5d, e). In $Nek1^{Kat2J/Kat2J}$ MEFs, the expression of VPS26B$^{S302D/S304D}$, but not VPS26B$^{WT}$ or VPS26B$^{S302A/S304A}$, promoted the localization of GLUT1 to the plasma membrane (Fig. 5f, g). Thus, the phosphorylation of VPS26B Ser302/Ser304 by NEK1 is important to shuttle GLUT1 from endosomes to the plasma membrane. Taken together, Nek1 deficiency may inhibit the recycling of SNX27-retromer from plasma membrane back to endosome which makes it unavailable to deliver additional GLUT1 from endosome to cell surface membrane.

We next analyzed the functional consequences of reduced plasma membrane presence of GLUT1 using $^{13}C_6$-labelled glucose uptake. We found that $Nek1^{Kat2J/Kat2J}$ MEFs showed dramatically reduced levels of $^{13}C_6$-glucose uptake as well as $^{13}C$-labelled downstream metabolites of $^{13}C_6$-glucose compared to WT MEFs in fast $^{13}C$-labelling experiment (Fig. 5h). Longer term $^{13}C$-labelling experiment also demonstrated the reduction in the levels of pyruvate in $Nek1^{Kat2J/Kat2J}$ MEFs (Supplementary Fig. 5c). With reduced precursor metabolites such as pyruvate from glycolysis, the TCA activity of $Nek1^{Kat2J/Kat2J}$ MEFs was also compromised, which were evidenced by the reduced oxygen consumption rate in seahorse analysis (Supplementary Fig. 5d), reduced levels of acetyl-coA (Supplementary Fig. 5e) and reduced levels of ATP (Supplementary Fig. 5f). Consistent with reduced glucose uptake and the activation of starvation response, $Nek1^{Kat2J/Kat2J}$ MEFs showed higher levels of p-S79 ACC, a substrate of AMPK, than that of WT (Supplementary Fig. 5g). The levels of p-S79 ACC in $Nek1^{Kat2J/Kat2J}$ and WT MEFs were elevated to the same levels when treated with metformin, which is known to activate AMPK[44] (Supplementary Fig. 5g). Treatment with Nec-1s had no effect on the levels of p-ACC (Supplementary Fig. 5h). Thus, AMPK is likely activated in $Nek1^{Kat2J/Kat2J}$ MEFs due to inadequate glucose uptake. These results suggest that NEK1 deficiency disrupts retromer-mediated trafficking which in turn impairs glucose uptake due to reduced GLUT1 in plasma membrane and leads to metabolic defects upstream of RIPK1.

Consistence with the expression of VPS26B$^{WT}$ and VPS26B$^{S302D/S304D}$ promoted the localization of GLUT1 to the plasma membrane (Fig. 5d, e), addback of VPS26B$^{WT}$ and VPS26B$^{S302D/S304D}$ significantly rescued the decreased glucose

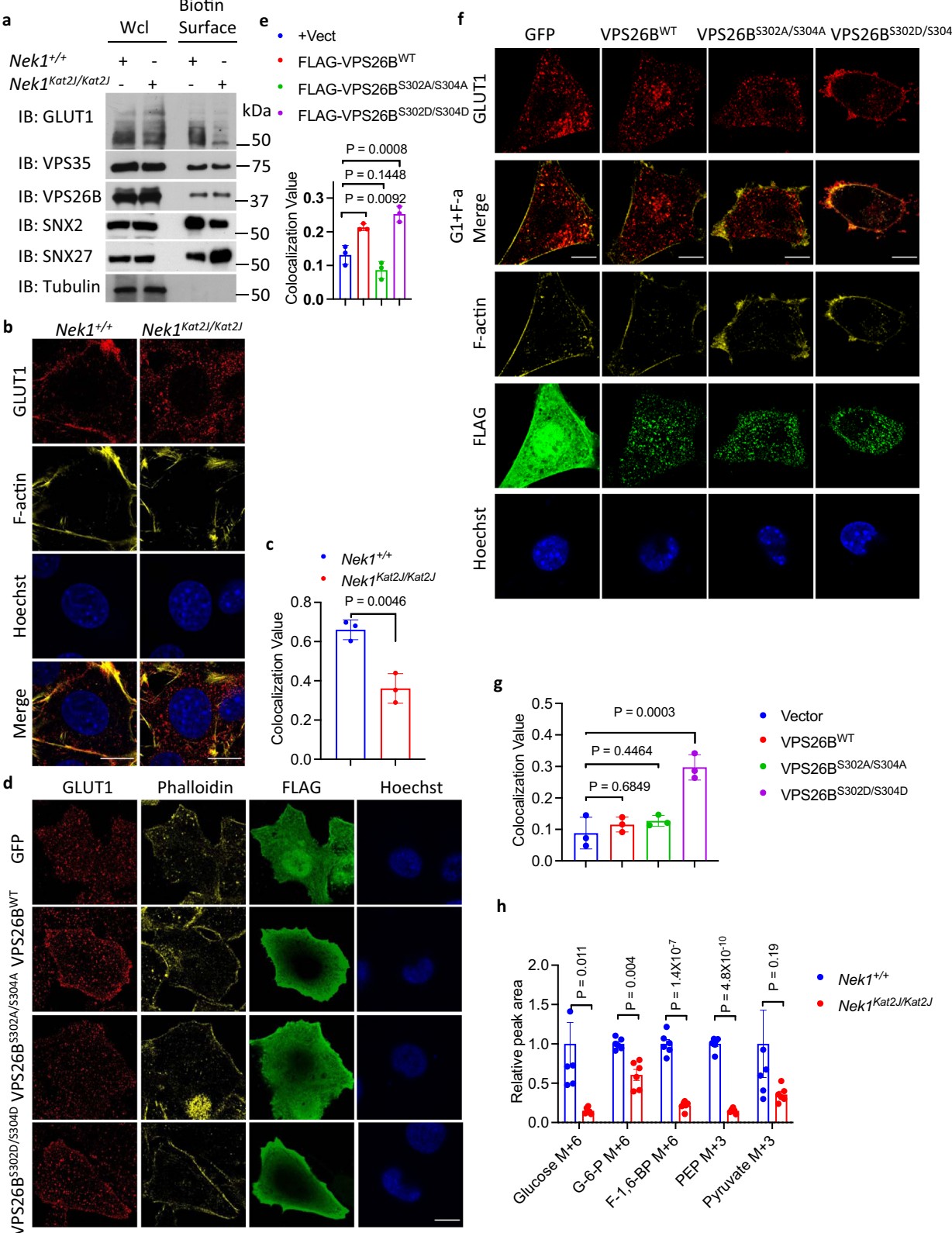

uptake and acetyl-CoA levels in VPS26B knockdown cells, while the addback of VPS26B$^{S302A/S304A}$ showed no significant effect on glucose uptake and acetyl-CoA levels in VPS26B knockdown cells (Supplementary Fig. 5i, j).

**Metabolic rescue of increased RIPK1 activation in *Nek1* deficient cells.** Given the defect in glucose uptake in *Nek1* deficient

cells, we considered whether cellular metabolism could be rescued by promoting fatty acid metabolism. ALCAR (acetyl-L-carnitine), the principal acetyl ester of L-Carnitine (LC), is a donor of acetyl groups and facilitates the transfer of fatty acids from cytosol to mitochondria to promote β-oxidation[45]. We found that ALCAR pretreatment reduced the sensitivity of *Nek1$^{Kat2J/Kat2J}$* MEFs to RDA and necroptosis, while effects on WT MEFs were minimal

**Fig. 5 NEK1 deficiency disrupts glucose uptake. a** $Nek1^{+/+}$ and $Nek1^{Kat2J/Kat2J}$ cells were treated with Sulfo-NHS-SS-Biotin to label cell-surface proteins in intact MEFs. The isolated plasma membrane proteins and whole cell lysates were analyzed by western blotting with indicated antibodies. Uncropped blots in the Source Data file. **b, c** $Nek1^{+/+}$ and $Nek1^{Kat2J/Kat2J}$ MEFs were immunostained with anti-GLUT1 antibody and also with Phalloidin for F-actin. The cell surface presence of GLUT1 was quantified using CellProfiler. Bar = 20 μm. Mean ± SD, $n = 3$ biological independent experiments. Two-tailed Student's t test (**c**). **d, e** HeLa cells were transfected with expression vectors for FLAG tagged VPS26B$^{WT}$ and VPS26B$^{S302A/S304A}$ for 24 h and immunostained with anti-GLUT1 antibody and Phalloidin. The cell surface presence of GLUT1 was quantified using CellProfiler, mean ± SD, n=3 biological independent experiments. Bar = 10 μm. One-way ANOVA with Dunnett's test (**e**). **f, g** $Nek1^{Kat2J/Kat2J}$ MEFs were transfected with indicated expression vectors for 24 h and immunostained with anti-GLUT1 and Phalloidin. Bar = 20 μm. The cell surface presence of GLUT1 was quantified using CellProfiler (mean ± SD). $n = 3$ biological independent experiments. One-way ANOVA with Dunnett's test (**g**). **h** $Nek1^{+/+}$ and $Nek1^{Kat2J/Kat2J}$ MEFs were treated with 4 mM U-$^{13}$C-glucose for 7 min and the cell lysates were analyzed by mass spectrometry for uptake and metabolism of glucose. The levels of $^{13}C_6$ labeled glucose, Glucose-6-phosphate and fructose-1,6-bisphosphotate and metabolic products of $^{13}C_6$ labeld glucose including $^{13}C_3$ labeled phosphoenolpyruvate, pyruvate were quantified. Data are presented as mean ± SEM (n=6 biological independent repeats). Unpaired multiple t-test.

(Fig. 6a; Supplementary Fig. 6a). Octanoate was known to drive fatty acid oxidation which may compensate for inadequacy in glucose metabolism[46]. We found that pretreating $Nek1^{Kat2J/Kat2J}$ MEFs with sodium octanoate effectively reduced sensitivity to RDA (Fig. 6b).

Retromer complexes regulate lysosomal functions by mediating the translocation of lysosomal hydrolases[47, 48]. We found that the lysosomes in $Nek1^{Kat2J/Kat2J}$ MEFs were much larger in sizes than that of WT MEFs as shown by immunofluorescent staining of LAMP1, a lysosome marker (Fig. 6c–e). The protein levels of LAMP1 were also increased in $Nek1^{Kat2J/Kat2J}$ MEFs compared to that of WT MEFs (Fig. 6f). $Nek1$ deficiency resulted in dramatically decreased lysosomal activity as determined by the cathepsin B activity assay using the Magic Red probe and increased protein levels of p62, which were not affected by the treatment with Nec-1s (Fig. 6f, Supplementary Fig. 6b). Consistent with increased levels of α-synuclein in the brains of $Nek1^{Kat2J/Kat2J}$ mice, $Nek1^{Kat2J/Kat2J}$ MEFs transfected with an expression vector of α-synuclein showed increased accumulation compared to WT MEFs (Supplementary Fig. 6c). ALCAR pretreatment showed no obvious effect on the cathepsin B activity assay using the Magic Red probe, protein levels of p62 and LAMP1, nor did it have any significant effect on the enlarged LAMP1$^+$ lysosome size in $Nek1^{Kat2J/Kat2J}$ MEFs (Supplementary Fig. 6d–f).

Normal retromer function is required for mitochondrial homeostasis[49, 50]. We found that $Nek1^{Kat2J/Kat2J}$ MEFs showed abnormal fragmentation of mitochondria (Fig. 6g). Treatment with ALCAR successfully rescued the morphology of mitochondria in $Nek1^{Kat2J/Kat2J}$ MEFs (Fig. 6h). Taken together, these results suggest that $Nek1$ deficiency promotes the defects in lysosomal activity and mitochondrial homeostasis. Increased fatty acid metabolism can ameliorate mitochondrial defects but not lysosomal dysfunction. Addback of VPS26B$^{WT}$ and VPS26B$^{S302D/}$ $^{S304D}$, but not VPS26B$^{S302A/S304A}$, reduced the enlarged lysosomes and mitochondria fragmentation in VPS26B knockdown cells (Supplementary Fig. 6g, h).

Glucose is an important source of acetyl-CoA for cells. Consistent with inadequate glucose uptake in $Nek1^{Kat2J/Kat2J}$ cells, we found the acetyl-CoA levels were also lower in $Nek1^{Kat2J/Kat2J}$ cells (Supplementary Fig. 5e). Since the levels of acetyl-CoA, an important donor of acetyl-group, can regulate the acetylation of A20 protein and reduced acetylation of A20 protein ptomotes its lysosomal degradation[51], we examined the levels of acetylated A20 in $Nek1^{Kat2J/Kat2J}$ cells and whether ALCAR and sodium octanoate could upregulate A20. Consistent with the decreased acetyl-CoA levels, the acetylation of A20 protein was much lower in $Nek1^{Kat2J/Kat2J}$ cells (Supplementary Fig. 6i). We found that treatment with ALCAR or sodium octanoate substantially rescued the A20 protein levels in $Nek1^{Kat2J/Kat2J}$ MEFs (Fig. 6i,

Supplementary Fig. 6j). Treatment with Nec-1s has no effects on the enlarged lysosomes, decreased glucose uptake into cells, decreased acetyl-CoA levels and mitochondrial fragmentation in $Nek1^{Kat2J/Kat2J}$ cells (Supplementary Fig. 6k–n). Consistent with restoration of A20 levels, pretreatment with ALCAR reduced the phosphorylation of necroptosis biomarkers, p-S166 RIPK1, p-T231/S232 RIPK3, and p-S345 MLKL in $Nek1^{Kat2J/Kat2J}$ MEFs (Fig. 6j). Thus, metabolic dysfunction in $Nek1^{Kat2J/Kat2J}$ cells reduces the protein levels of A20 protein by reducing its acetylation which in turn promotes its degradation to sensitize to RIPK1 activation and necroptosis.

**Ketogenic diet rescues the BBB damage and postnatal lethality of $Nek1^{Kat2J}$ mice.** Phosphorylation of S79 in ACC (acetyl-CoA carboxylase) by AMPK is a biomarker for cellular starvation response[52]. Supporting the defect of glucose uptake induced by $Nek1$ deficiency, we found that the brain lysates of $Nek1^{Kat2J/Kat2J}$ mice contained higher levels of p-S79 ACC than that of WT mouse brains which were not affected by inhibition of RIPK1 in $Nek1^{Kat2J/Kat2J}$; $Ripk1^{D138N/D138N}$ mice (Fig. 7a). Ketogenic diet is used clinically for treatment of patients with Glut1 Deficiency[53]. Thus, we examined if a ketogenic diet might be able to rescue the metabolic defects caused by $Nek1$ deficiency. We administered a ketogenic diet to $Nek1^{Kat2J/+}$ breeders from the beginning of the mating as well as their $Nek1^{Kat2J/Kat2J}$ progeny. In contrast to substantial amount of extravasated Evans blue in the brains of $Nek1^{Kat2J/Kat2J}$ mice on control diet, $Nek1^{Kat2J/Kat2J}$ newborns from mothers who had been on ketogenic diet since pregnancy showed no extravasated Evans blue (Fig. 7b, Supplementary Fig. 7a). Ketogenic diet also inhibited necroptosis and RDA of cerebrovascular endothelial cells in the brains of $Nek1^{Kat2J/Kat2J}$ mice; no obvious phosphorylation of MLKL or cleaved caspase-3 were observed in the brains of $Nek1^{Kat2J/Kat2J}$ mice on ketogenic diet (Supplementary Fig. 7b, c). Ketogenic diet also extended the survival of $Nek1^{Kat2J/Kat2J}$ mice; of 7 $Nek1^{Kat2J/Kat2J}$ mice on ketogenic diet, one lived until the age of ~9 weeks, the remaining 6 mice survived more than 5 months until they had to be euthanized due to malocclusion caused by craniofacial abnormality at the age of 5 to 11 months (Fig. 7c). We examined the numbers of Gr1$^+$ neutrophils in $Nek1^{Kat2J/Kat2J}$ mice fed with control or ketogenic diet and found that ketogenic diet reduced the numbers of Gr1$^+$ neutrophils in the cerebral cortex and thalamus of $Nek1^{Kat2J/Kat2J}$ mice (Fig. 7d, e). Ketogenic diet suppressed the increase in p-ACC and substantially normalized the levels of A20 protein in the cerebrovasculature of $Nek1^{Kat2J/Kat2J}$ mice (Fig. 7f, g). In addition, we compared the mRNA levels of elevated proinflammatory factors in the brains and spinal cords of $Nek1^{Kat2J/Kat2J}$ mice that were fed with control and ketogenic diet, including $Tnf$, $Il1\alpha$, $Il1\beta$, $Il6$, $Cxcl1$, $Ccl2$, $Ccl5$, and $Cst7$, and found that their mRNA levels were reduced by ketogenic diet (Fig. 7h, i). The

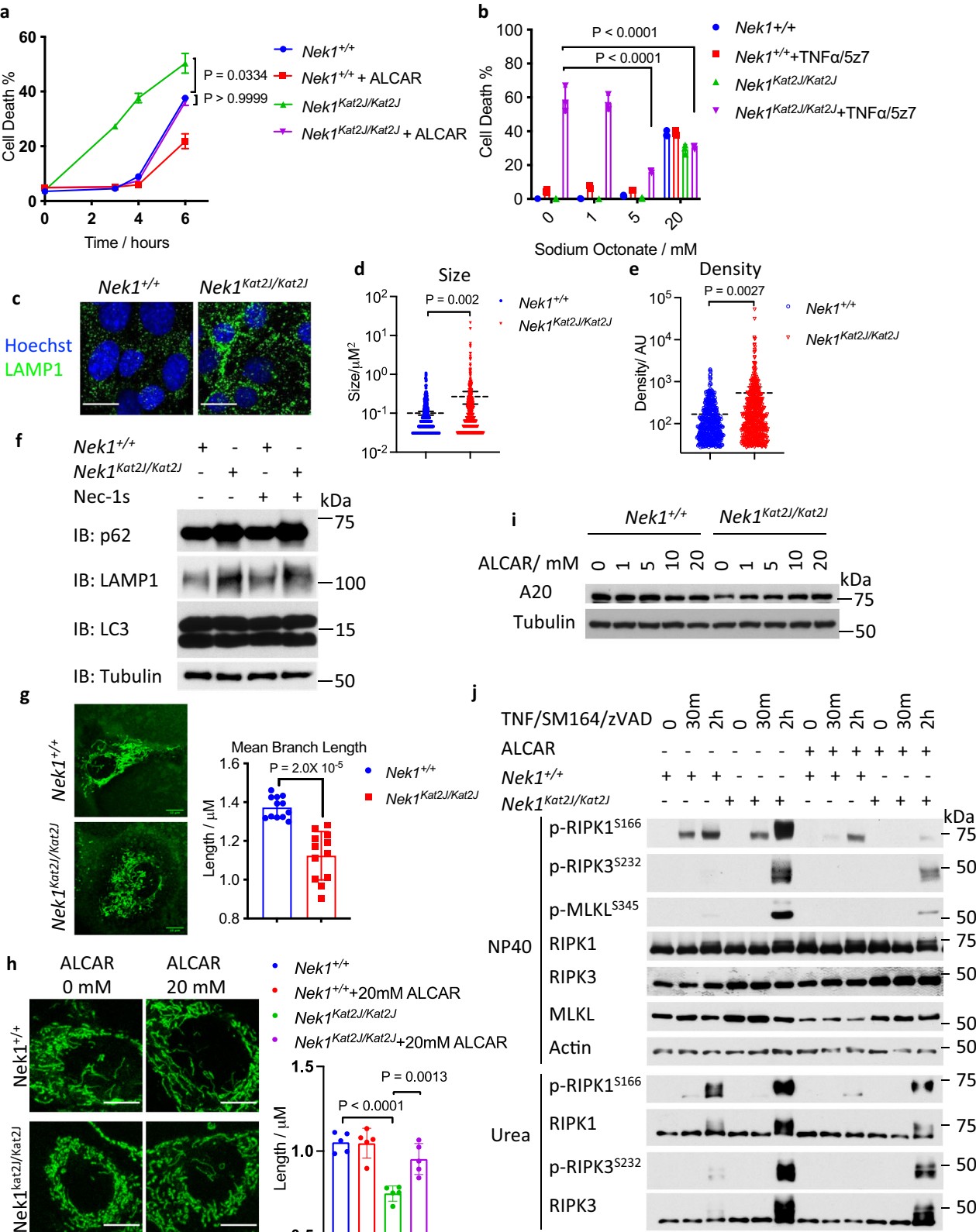

ketogenic diet showed no significant effects on body weight of WT and *Nek1*$^{Kat2J/Kat2J}$ mice compared to the control diet (Supplementary Fig. 7d, e). Taken together, we conclude that the the preweaning lethality of *Nek1*$^{Kat2J/Kat2J}$ mice can be reduced by inhibition of RIPK1 as well as by ketogenic diet which can rescue metabolic defects to inhibit RIPK1 mediated cell death and inflammation (Supplementary Fig. 7f).

## Discussion

Our data demonstrates the mechanism of NEK1 in regulating retromer-mediated endosomal trafficking by phosphorylating VPS26B, a component of the retromer complex[54]. Our results suggest that the phosphorylation of S302/S304 VPS26B mediated by NEK1 faciliates the binding with SNX2 but is prohibitive to the binding with SNX27. Thus, by regulating VSP26B

**Fig. 6 Metabolic rescue of increased RIPK1 activation in NEK1 deficient cells. a** $Nek1^{+/+}$ and $Nek1^{Kat2J/Kat2J}$ MEFs were pre-treated with 20 mM acetyl-l-carnitine for 18 h and then treated with 1 ng/mL TNF + 500 nM 5z7 to induce RDA for 3, 4, and 6 h. Cell death was measured by ToxiLight. Mean ± SD, n=3 biological independent samples. One-way ANOVA with Dunnett's test. **b** $Nek1^{+/+}$ and $Nek1^{Kat2J/Kat2J}$ MEFs were pre-treated with indicated concentrations of sodium octonate for 18 h and then treated with 1 ng/mL TNF + 500 nM 5z7 to induce RDA for 6 h. The % of cell death was measured by ToxiLight. Mean ± SD. $n = 3$ biological independent repeats. One-way ANOVA with Dunnett's test. **c–e** The sizes of lysosomes in $Nek1^{+/+}$ and $Nek1^{Kat2J/Kat2J}$ MEFs were determined with a biomarker LAMP1 by immunostaining (**c**) and quantified with ImageJ for sizes (**d**) and density (**e**). $n = 6$ cells per group. Bar = 10 μm. Data are presented as mean ± SEM. Unpaired two-tailed Student's t-test. **f**, Whole-cell lysates of $Nek1^{+/+}$ and $Nek1^{Kat2J/Kat2J}$ MEFs were analyzed by western blotting with indicated antibodies. Uncropped blots in the Source Data file. **g** $Nek1^{+/+}$ and $Nek1^{Kat2J/Kat2J}$ MEFs were stained with 50 nM MitoTracker-Green for 30 min and imaged by fluorescent microscopy. The length of mitochondria was quantified with ImageJ. Mean ± SD, $n = 12$ cells for each group. Bar = 10 μm. Unpaired two-tailed Student's t-test. **h**, $Nek1^{+/+}$ and $Nek1^{Kat2J/Kat2J}$ MEFs were treated with indicated concentration of acetyl-l-carnitine for 24 h, and then treated with 50 nM MitoTracker-Green for 30 min. The images were acquired by confocal microscopy. The average branch lengthes of mitochondria were quantified with ImageJ ($n = 5$ cells per group). The data are presented as mean ± SD. One-way ANOVA with Dunnett's test. Bar = 10 μm. **i** WT and $Nek1^{Kat2J/Kat2J}$ MEFs were treated with 0, 1 mM, 5 mM, 10 mM, and 20 mM acetyl-l-carnitine for 20 hours, then lysed with SDS buffer. Uncropped blots in the Source Data file. **j** $Nek1^{+/+}$ and $Nek1^{Kat2J/Kat2J}$ MEFs were pre-treated with 20 mM acetyl-l-carnitine for 18 h and then treated with 10 ng/mL TNF + 50 nM SM164 + 20 μM zVAD.fmk for indicated periods of time. The cell lysates lysed with NP40 buffer and with 6 M urea were analyzed by western blotting using indicated abs. Uncropped blots in the Source Data file.

phosphorylation, NEK1 may control the flux and direction of intracellular retromer trafficking. Blocking of S302/S304 VPS26B phosphorylation in NEK1 deficient cells inhibits the recycling of SNX27- and SNX2-retromers which distorts the cellular distribution of the retromer components and substrates that include increased plasma membrane levels of SNX27 and decreased plasma membrane levels of SNX2, as well as at least 79 other known substrates of the retromer, such as the critical glucose receptor GLUT1 (Fig. 8). Retromer-mediated intracellular trafficking is important in maintaining normal cellular homeostasis by mediating the recycling of nutrient transporters, mitogenic signaling receptors, cell adhesion receptors, and synaptic receptors from endosomes to the plasma membrane[13, 55], which explains why $Nek1$ deficiency leads to pleiotropic defects at the cellular level, including mitochondrial dysfunction and lysosomal defects, as well as systemic malformation during development, including BBB damage-mediated by RIPK1 activation, and the pre- and perinatal lethality of $Nek1^{Kat2J/Kat2J}$ mice. While $Nek1$-$Kat2J/Kat2J$ mice present the null-phenotype of $Nek1$, reduction-of-function mutations and missense mutations have been found in human ALS[56]. Since the early lethal phenotype of $Nek1^{Kat2J/Kat2J}$ mice precludes the analysis of neurological dysfunction caused by $Nek1$ deficiency, it will be interesting to examine the impact of such partial loss of NEK1 function on the development of neurological defects in future.

Dysregulation of ubiquitination has been implicated in mediating neurodegeneration[57]. A20 is an important ubiquitin editing enzyme implicated in multiple human inflammatory diseases[25, 26]. Cell lineage-specific deficiency of A20 in mice recapitulates several human inflammatory diseases involving RIPK1[26]. Here we show that metabolic dysfunction in $Nek1$ deficient cells reduces the levels of A20 to promote the activation of RIPK1, necroptosis of CD31+ endothelial cells, and BBB damage, which is critical for the postnatal lethality of $Nek1^{Kat2J/Kat2J}$ mice. Our study reveals the mechanism by which dysfunction in metabolism may promote the activation of RIPK1: unlike ALS-associated genetic variants of OPTN and TBK1, which directly promote the activation of RIPK1[58, 59], NEK1 can indirectly modulate the activation of RIPK1 through regulating retromer trafficking and metabolism which in turn controls the levels of A20 by regulating the levels of acetyl-CoA. Consistently, ketogenic diet, which may provide alternate source of acetyl-CoA, can also suppress the activation of RIPK1, neuroinflammation and early lethality in $Nek1^{Kat2J/Kat2J}$ mice. Thus, the activation of RIPK1 in $Nek1$ deficient cells is a downstream consequence of metabolic dysfunction caused by defects in retromer. Consistently, inhibition of RIPK1 has no effect on the other

cellular defects caused by retromer dysfunction including lysosome which is known to be regulated by retromer[60]. Our results suggest that the metabolic defects in cerebral endothelial cells in different disease conditions might sensitize to the activation of RIPK1 due to reduction in the levels of A20.

Our study highlights the role of RIPK1 and necroptosis of cerebrovascular endothelial cells in mediating CNS pathology. We have shown that the necroptosis of cerebromicrovascular endothelial cells is involved in mediating BBB damage in Alzheimer's disease and ischemic stroke[51, 61]. Here we show the role of RIPK1-mediated necroptosis and apoptosis of cerebromicrovascular endothelial cells in BBB damage due to NEK1 deficiency. Necroptosis and RIPK1-mediated inflammation has been implicated in mediating neurodegenerative diseases such as ALS associated with specific mutations in TNFR1 pathway including OPTN and TBK[58, 59, 62–65]. Our study suggests that inhibition of RIPK1 may reduce neuroinflammation and the accumulation of misfolded proteins in neurodegenerative diseases that are not genetically connected with mutations in the genes that directly regulate the activation of RIPK1. Inhibition of RIPK1 not only genetically rescued the postnatal lethality of $Nek1^{Kat2J/Kat2J}$ mice by inhibiting BBB damage, but also reduced neuroinflammation and the accumulation of misfolded proteins such as α-synuclein, suggesting that the activation of RIPK1-mediated inflammation might also contribute to the disruption of cellular homeostasis which in turn promotes the accumulation of α-synuclein in the pathogenesis of Parkinson's diseases and the cognitive dysfunction in Alzheimer's disease[66–68].

The therapeutic value of a ketogenesis such as by following a Mediterranean diet has been investigated as a metabolic supplement for neurodegenerative diseases[69]. Impairment of glucose uptake and defective glycolysis has been noted in ALS[70]. Our study highlights the role of acetyl-CoA as a metabolic regulator of RIPK1. Acetyl-CoA is mainly produced by the breakdown of carbohydrates through glycolysis and lipids through β-oxidation. In low glucose uptake, the origin of Acetyl-CoA heavily relies on lipid β-oxidation, which could be facilitated by ketogenic diet. Other origins of Acetyl-CoA such as from acetate and amino acids might also contribute. Altered fuels, such as ketogenic diet with increased components of C4 ketones, the addition of ketogenic medium-chain triglyceride trioctanoin, the triglyceride of octanoate (caprylic acid), have been suggested as a metabolic supplement for ALS patients. Ketogenic diet has been shown to increase motor function and survival in a mouse model of ALS (SOD1[G93A])[71]. Our study supports the use of ketogenic diet in a sub-population of ALS patients with metabolic defects to provide additional acety-CoA which may indirectly suppress the

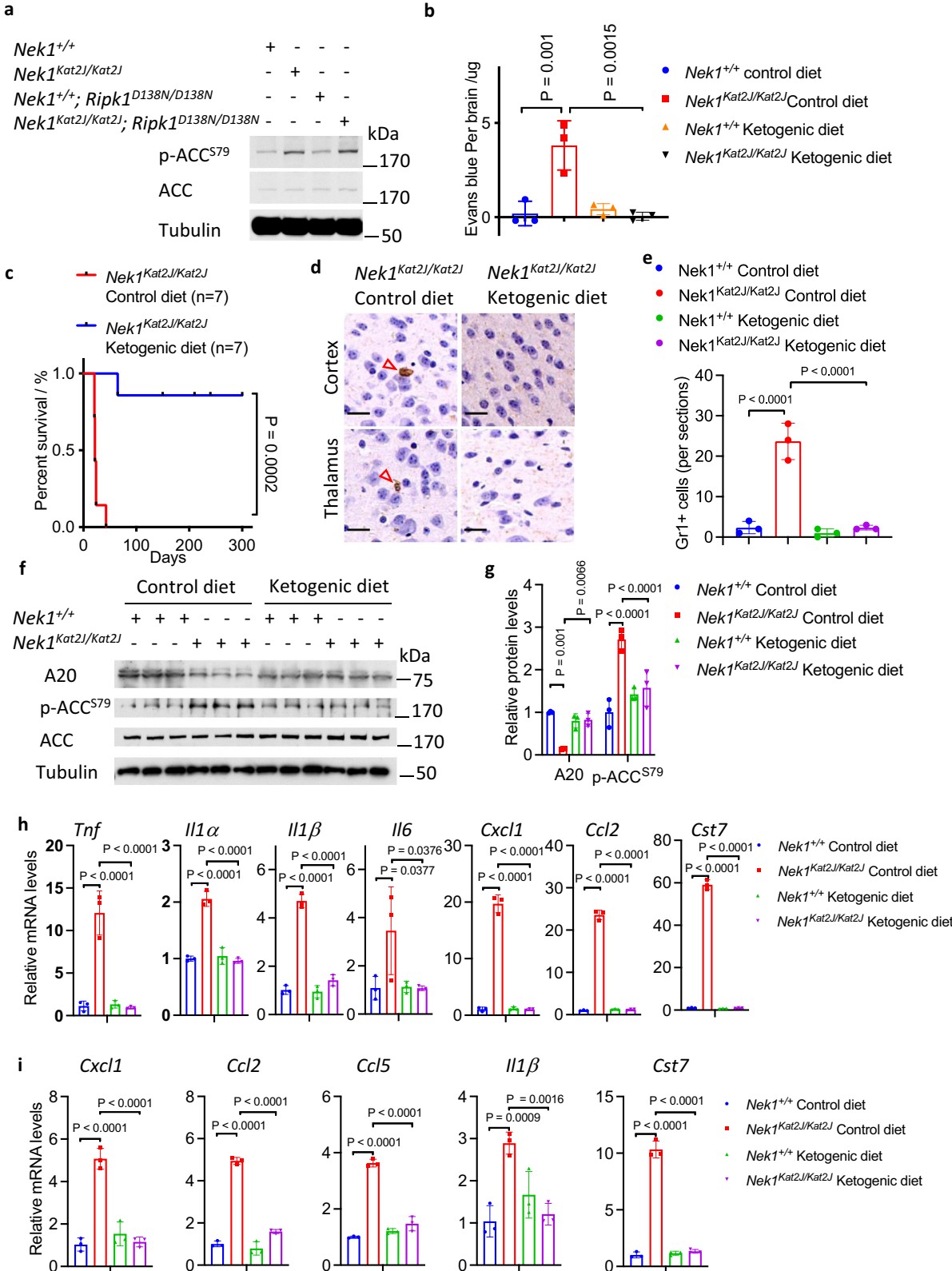

activation of RIPK1 by regulating the levels of A20 to reduce neuroinflammation and accumulation of misfolded proteins.

Our study highlights the role of the retromer in regulating homeostasis of the CNS and reveals how retromer dysfunction may mediate neurodegeneration. We show that metabolic defects in *Nek1^Kat2J* mice promote α-synuclein aggregation, a hallmark of PD, which can be ameriorated by inhibition of RIPK1. Genetic

variants of retromer components have been associated with AD and PD such as *VPS35* genetic variants[72]. A gene expression profiling study has found that VPS26 is downregulated in AD and implicated in promoting Aβ production[73]. Variants of the *VPS35* gene (also known as *PARK17*) have been recognized as a cause of late-onset familial PD[74]. Defects in retromer-mediated endosomal trafficking promote lysosomal dysfunction and disrupt

**Fig. 7 Ketogenic diet rescues the BBB damage and preweaning lethality of *Nek1^Kat2J* mice. a** Whole brain lysates of *Nek1^+/+* mice, *Nek1^Kat2J/Kat2J* mice, *Nek1^+/+;Ripk1^D138N/D138N* mice and *Nek1^Kat2J/Kat2J;Ripk1^D138N/D138N* mice were analyzed by western blotting with indicated antibodies. Uncropped blots in the Source Data file. **b** The integrity of BBB in *Nek1^+/+* newborn mice and *Nek1^Kat2J/Kat2J* newborn mice, fed with control diet or ketogenic diet since the pregnancy of their mothers, was assessed by intraperitoneal injection of Evans blue and the cerebral retention of Evans blue was assessed at 24 h after the injection with PBS perfusion. $n = 3$ mice for each group. One-way ANOVA with Dunnett's test. **c** The postnatal survival of *Nek1^Kat2J/Kat2J* mice on control diet or ketogenic diet. $N = 7$ for each group. Unpaired two-tailed Student's t-test. **d, e** Gr1 immunohistostaining (brown) of cerebral cortex and thalamus sections from the brains of *Nek1^Kat2J/Kat2J* or WT mice (3 weeks old) fed with ketogenic diet or control diet (**d**) and the numbers of Gr1 positive cells per sections were quantified (**e**). Bar = 20 μm. $n = 3$ sections per group. One-way ANOVA with Dunnett's test. **f–g** Cerebromicrovessles were isolated from the brains of 3 *Nek1^+/+* mice and 3 *Nek1^Kat2J/Kat2J* mice fed with control diet or ketogenic diet. The lysates of isolated cerebromicrovessles of each genotypes were analyzed by western blotting with indicated antibodies (**f**) and quantified using ImageJ (**g**). $n = 3$ replicates for each group. One-way ANOVA with Dunnett's test. Uncropped blots in the Source Data file. **h-i** The mRNAs isolated from brains (**h**) and spinal cords (**i**) of *Nek1^+/+* mice and *Nek1^Kat2J/Kat2J* mice fed with control diet or ketogenic diet (n=3 per group, 3 weeks old) were quantified for the levels of indicated cytokines by qPCR. Data are presented as mean ± SD. One-way ANOVA with Dunnett's test.

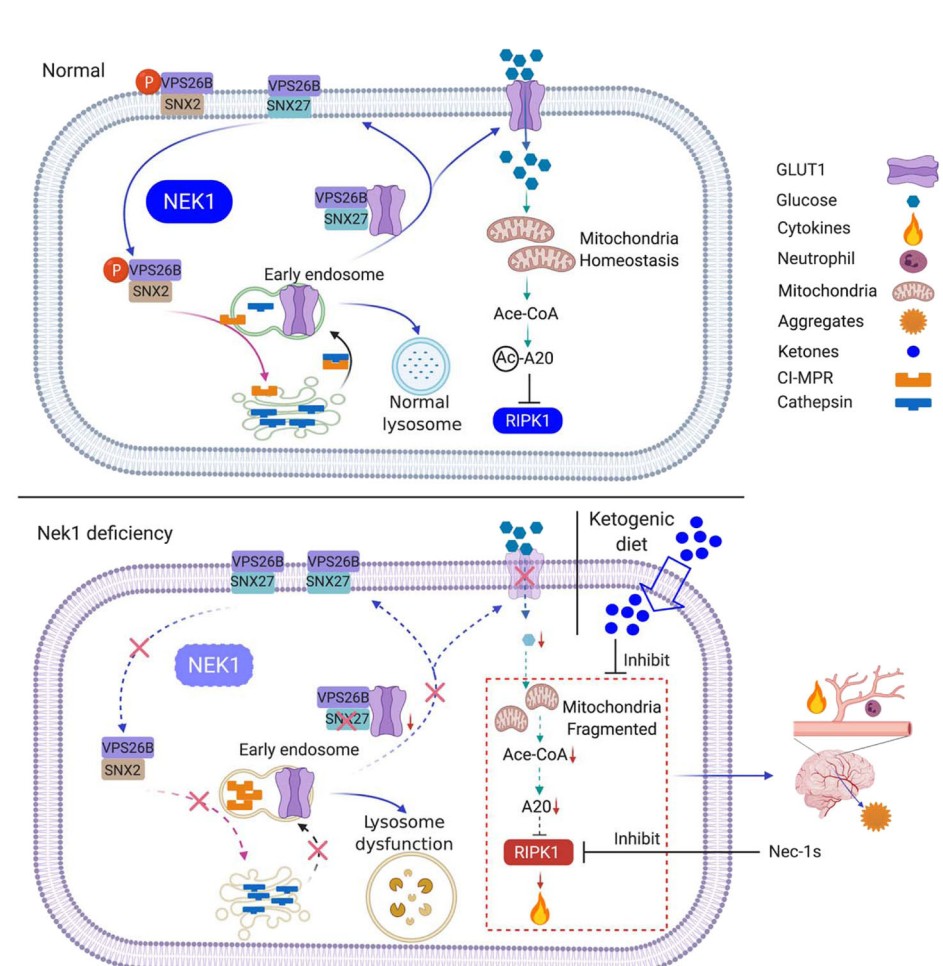

**Fig. 8 A model for NEK1-regulated retromers that control the cell surface presence of membrane receptors, mitochondrial metabolism and RIPK1-mediated cell death and inflammation.** Phosphorylation of S302/S304 VPS26B by NEK1 modulates the recycling of SNX27-retromer and SNX2-retromer that deliver proteins from endosomes to cell surface membrane and lysosomes. NEK1 deficiency blocks the recycling of SNX27-retromer from cell surface to endosomes. Since SNX27-retromer is important for delivering GLUT1 to the cell surface membrane, NEK1 deficiency leads to reduced GLUT1 levels at the cell surface membrane which reduces glucose uptake. Reduced glucose uptake disrupts mitochondrial metabolism and reduces cellular levels of acetyl-CoA. Reduced acetyl-CoA inhibits the acetylation of A20 which promotes its lysosomal degradation. Reduced levels of A20 sensitizes to the activation of RIPK1-dependent cell death and inflammation which can be inhibited by Nec-1s and also by ketogenic diet which provides alternate source of acetyl-CoA to restore cellular levels of A20.

mitochondrial homeostasis[50]. Thus, defective retromer-mediated endosomal trafficking may be involved in the pathogenesis of AD and PD, but also ALS as shown in this study. Although it remains unclear how genetic variants of retromer components may promote the onset of distinct pathology in different

neurodegenerative diseases, our study suggests that the regulation of different subtypes of retromer complexes, such as the phosphorylation of VSP26B by NEK1, and the control of subpopulations of retromer substrate trafficking may contribute to the specificity of neuropathology. Overall, our study suggests the

possibility of inhibiting RIPK1 kinase for the treatment of ALS, AD, and also PD, to prevent cell death resulting downstream of myriad etiologies of neurodegeneration, but also suggests that metabolic regulations may provide additional benefit for therapeutic intervention.

## Methods

**Animals.** C57BL/6J-Nek1$^{kat-2J}$/J mice were purchased from The Jackson Laboratory (stock number: 002854). Nek1$^{Kat2J/+}$ mice were crossed with Ripk1$^{D138N/D138N}$ mice to generate Nek1$^{Kat2J/+}$; Ripk1$^{D138N/+}$ mice. Nek1$^{Kat2J/+}$;Ripk1$^{D138N/+}$ mice were crossed with Ripk1$^{D138N/D138N}$ mice to generate Nek1$^{Kat2J/+}$; Ripk1$^{D138N/D138N}$ mice. All animals were maintained in a pathogen-free environment, and experiments on mice were conducted according to the protocol IS00001127 approved by the Harvard Medical School Animal Care Committee on Animal Care. Mice were group-housed with free access to water and food and in a 12 hr light/dark cycle (light between 07:00am and 7:00 pm) in a temperature-controlled room 20.5 °C) at Harvard Medical School. The ketogenic diet was purchased from Bio-Serv company, cat#S3666. The default chow for all other mice is from LabDiet Company, cat#5053.

**Evans blue extravasation assay.** For Evan's blue extravasation assays, P21 Nek1$^{Kat2J/Kat2J}$ mice, Nek1$^{Kat2J/Kat2J}$; Ripk1$^{D138N/D138N}$ mice, and littermate wild type mice were given an intraperitoneal injection of 2% Evan's blue in PBS solution (70 mg/kg). Twenty-four hours after injection of Evans blue, animals were anesthetized with isoflurane, and then perfused with PBS followed by 4% paraformaldehyde for image analysis or with PBS only for quantification by homogenizing each brain in 2 mL 50% TCA. Samples were centrifuged at 17000 g for 15 min, then 80 μL of each supernatant were added to a 96-well-plate with 3 repeats for each sample. Evans blue standard solutions were used: 0, 0.2 ug/mL, 0.5 ug/mL, 1 ug/mL, 5 ug/mL, 10 ug/mL and 80 μL with 3 repeats of each standard solution. Absorption at 620 nm wavelength was used to measure concentration of samples. The intensity of absorption was determined in an EnSpire Multimode Plate Reader (PerkinElmer). Data were collected using PerkinElmer EnVision Manager Version 1.13 software.

**Generation and immortalization of MEFs.** WT, Nek1$^{Kat2J/+}$ and Nek1$^{Kat2J/Kat2J}$ embryonic fibroblast cells were isolated from littermate E11.5-13.5 embryos produced by mating of Nek1$^{Kat2J/+}$ female mice (8-10 weeks) with Nek1$^{Kat2J/+}$ male mice (8-10 weeks old). Primary MEFs were cultured in high glucose DMEM supplemented with 15% FBS, non-essential amino acids, sodium pyruvate, penicillin and streptomycin. MEFs were immortalized by transfection with SV40 large T antigen expressing plasmid (Addgene Cat#22298) using Lipofectamine 3000.

**Primary microglia culture.** Forebrains of sacrificed 1-day-old mouse pups were digested with 0.01% trypsin and triturated with DMEM containing 10% heat-inactivated fetal bovine serum and 1% penicillin-streptomycin. Dissociated cells were plated onto poly-d-lysine coated 75 cm$^2$ flasks and the media was changed every 7 d for culturing 2 weeks in total. Microglia were collected by shaking the culture for 1 h and subsequently cultured in DMEM+10%FBS. The quality of purification was analyzed by FACS using CD11b as a marker for microglia (> 90% CD11b+).

**Plasmids construction.** Full length CDS domain of mouse NEK1 and VPS26B were PCR amplified from cDNA generated from mRNA of wild type MEFs, cDNA encoding HA tagged full length mouse NEK1, FLAG tagged full length mouse VPS26B and VPS26B$^{aa1-297}$ were cloned into pLenti vector for mammalian expression. D146N, C3107G, R261H murine NEK1 mutants and S311D, S311A, S302A/S304A, S302D/S304D murine VPS26B mutants were generated using MutExpress II mutagenesis kit (Vazyme Biotech Co., Ltd). The shRNA sequences used in this study were selected from the RNAi Consortium (TRC) (Broad Institute, MA, USA), the annealed shRNA oligonucleotides were cloned into the pLKO.1 vector. All plasmids were verified by DNA sequencing.

**Lentivirus production and infection.** pLenti based vectors for mammalian expression of HA tagged NEK1, NEK1$^{D146N}$, NEK1$^{C3107G}$, NEK1$^{R261H}$, FLAG tagged VPS26B, VPS26B$^{S302A/S304A}$, VPS26B$^{S302D/S304D}$ were transfected into 293 T cells individually with VSVG, REV and MDL packaging vectors for 8 hours and then cell medium was replaced with fresh medium. PLKO.1 based shRNA targeting mouse Nek1 (5'- CCATCAGTCAACTCCATATTG −3', 5'- AGCCGAAGGACACGTGGTTTA −3', 5'- CGCCAACAGATTAAAGCCAAA −3'), Vps26a (5'- CAATCGCTAAG-TATGAAATAA −3', 5'- CCTGATGTCAACAACTCTATT −3'), Vps26b (5'- ACGACACAATAGCGAAGTATG −3', 5'- GGCAGATCGAACTCTACTATG −3', 5'- TGCAGCCTAGGTAGGGATAAG−3'), Vps29 (5'-GCTGGTCTCCACCACT-TATTT−3', 5'-GTAGAACGAATTGAGTATAAA−3') or a scrambled sequence (5'- CAACAAGATGAAGAGCACCAA −3') were transfected into 293 T cells with VSVG, REV and MDL plasmids individually for 8 h and then cell medium was replaced with fresh medium. The cultural supernatant was harvested from transfected 293 T cells at 48 h after transfections and filtered with 0.45 μm membrane. Filtered medium

containing lentivirus particles was used to infect cells with Polybrene supplemented to the viral supernatant fractions at a final concentration of 8 μg/mL.

**Cell culture.** All cells were cultured at 37 °C with 5% CO2. L929 cells, RGC5 cells and HEK293T cells were originally obtained from the American Type Culture Collection (ATCC). MEFs were immortalized by infection of lentivirus expressing SV40T. MEFs, L929 cells, RGC5 cells and HEK293T cells were cultured in Dulbecco's modified Eagle's medium (DMEM; Thermo Fisher Scientific, cat. no. 11965) with 10% (vol/vol) fetal bovine serum (FBS; Thermo Fisher Scientific, cat. no. 10082-147). The cells were tested every two months using a TransDetect PCR Mycroplasma Detection Kit (Transgen Biotech, cat. no. FM311-01) to ensure that they were mycoplasma free.

**Western blotting.** Total cell lysates, isolated plasma membrane proteins or immunoprecipitated proteins were separated on SDS-PAGE and then transferred onto nitrocellulose membranes. Detection was performed with HRP-conjugated secondary antibodies. Antibodies against the following proteins were used for western blot analysis: NEK1 (Abcam #ab229489, 1:1000), NEK1 (Santa Cruz #sc-398813, 1:500), p-S166 RIPK1 (CST #31122, 1:1000), p-S166 RIPK1 (Arigo # ARG66476, 1:1000), RIPK1 (CST #3493, 1:2000), A20 (CST, #5630, 1:1000), TNFR1 (CST #13377, 1:1000), IκBα (CST #9242, 1:2000), p-S32/36 IκBα (CST, #9246, 1:2000), FLAG (Sigma-Aldrich #F3165, 1:2000), HA (CST #3724, 1:2000), α-synuclein (Invitrogen #AHB0261, 1:2000), p-S345 MLKL (CST #91689, 1:1000), MLKL (CST #37705, 1:1000), p-T231/S232 RIPK3 (CST #57220, 1:1000), RIPK3 (CST #13526, 1:1000), p-S79 acetyl-CoA carboxylase (CST #3661, 1:1000), acetyl-CoA carboxylase (CST #3662, 1:1000), SQSTM1/p62 (CST #5114, 1:5000), LAMP1 (Santa Cruz #sc-19992, 1:5000), p-Ser(EMD Millipore #05-1000, 1:2000). The signals were detected using Amersham ECL Western Blotting Detection Reagent (RPN2106, GE Healthcare) or homemade ECL.

**Cell surface protein isolation.** The isolation of cell surface proteins was conducted by following the manufacturer's protocol (Pierce™ Cell Surface Protein Isolation Kit, #89881, Thermo Scientific). Briefly, MEFs were seeded in 150 mm dishes at the density of 3 million cells per dish overnight. Six dishes were prepared for each sample. One vial of Sulfo-NHS-SS-Biotin was dissolved in 96 mL of ice-cold PBS. MEFs were washed with ice cold PBS quickly and 15 mL of the biotin solution was added to each dish. Plates were placed on orbital shaker and gently agitated for 30 min at 4 °C. Quenching solution (0.37 mL) was added to each dish to quench the reaction after the agitation. Cells were scraped and centrifuged down at 500 × g for 3 min and supernatant was discarded. Cells were washed with ice cold TBS. Lysis buffer supplemented with proteases inhibitors was added to cells and sonicated on ice to lyse the cells. Cell lysates were centrifuged at 10,000 × g for 2 min at 4 °C. The supernatant was collected and incubated with prewashed NeutrAvidin Agarose at room temperature with end-over-end mixing using a rotator for 60 min. The NeutrAvidin Agarose was then washed with washing buffer supplemented with proteases inhibitors and eluted with SDS-PAGE sample buffer supplemented with 50 mM DTT. Eluted proteins were precipitated with TCA and subjected to quantitative mass spectrometry based proteomic analysis.

**Analysis of cell death and viability.** Cell death was measured in triplicate or quadruplicate in a 96-well-plate by using ToxiLight Non-destructive Cytotoxicity BioAssay Kit (Lonza, LT07-117). The intensity of luminescence was determined in an EnSpire Multimode Plate Reader (PerkinElmer). The percentage of cell death per well was calculated by comparing to that of the maximal cell death with 100% Lysis Reagent after deducting background signal in non-induced cells. The rates of cell viability were determined by using CellTiter-Glo® Luminescent Cell Viability Assay (Promega, G7570) following the manufacturer's protocol and the results are expressed as percentages of luminescence intensity per well after deducting the background signal in blank well and compared to that of the viability in vehicle treated wells.

**Complex-I isolation and tandem IP of complex-I.** Cells were seeded in 150 mm dishes at the density of 4 million cells per dish overnight. MEFs were treated with 50 ng/mL FLAG-TNFα for indicated periods of time, then the medium was removed, and plates were washed with 50 ml of ice-cold PBS. Cells were lysed with NP40 buffer (50 mM Tris-HCl pH 7.5, 150 mM NaCl and 0.5% NP-40) supplemented with protease inhibitors and N-ethylmaleimide (2.5 mg/ml). The cell lysates were immunoprecipitated with anti-FLAG antibody conjugated agarose (1$^{st}$ IP). The immunoprecipitated proteins were eluted with 6 M urea buffer. Then the eluted proteins were diluted into 3 M urea buffer and immunoprecipitated with anti-K63 ubiquitin antibody (2$^{nd}$ IP). The whole lysis, eluted proteins from 1$^{st}$ IP and 2$^{nd}$ IP were subjected to SDS page and analyzed by western blotting with indicated antibodies.

**[U]$^{13}$C6-glucose labelling.** MEFs were plated in 6 cm dishes at 2,000,000 cells/dish and cultured in DMEM medium containing dFBS (10%) and penicillin/streptomycin (1%). MEFs were first grown to 75–80% confluence in log-phase in the cell culture plates. The culture medium was changed to a fresh medium with 4

mM [U]$^{13}$C6-glucose and incubated for 7 min for fast $^{13}$C labeling or 2 h for longer term $^{13}$C labeling. The [U]13C6-glucose culture medium was then removed, and the cells were washed with the cold PBS twice. The plates were placed on dry ice and the metabolite extraction solution (acetonitrile/methanol/water=2/2/1, v/v/v, 1 mL) was added to quench the metabolism. The extraction solution was pre-cooled at −80 °C for 1 h prior. The plates were then incubated at −80 °C for at least 40 min. The cell contents were scraped and transferred to a 1.5-mL Eppendorf tube. The samples were vortexed for 1 min, then spined down for 10 min at 17,000 g, 4 °C. The supernatant was taken to a new 1.5-mL Eppendorf tube, and evaporated to dryness at 4 °C using a vacuum concentrator. The dried extracts were then reconstituted in 100 μL of ACN:H2O (1:1, v/v), sonicated for 10 min, and centrifuged for 15 min at 17,000 g 4 °C to remove insoluble debris. The supernatant was then transferred to HPLC vials and kept at −80 °C until LC-MS analysis. The LC-MS analysis was performed using a UHPLC system (1290 series; Agilent Technologies, USA) coupled to a quadruple time-of-flight mass spectrometer (TripleTOF 6600, AB SCIEX, USA).

**Oxygen consumption rate assay.** Oxygen consumption rate (OCR) of cells were measured by using a Seahorse Biosience XF96 extracellular flux analyzer. 10000 cells/well were seeded in the XF96 plate overnight. The culture medium was changed with base medium 1 h prior to the assay, followed by sequential injection of oligomycin( uM), FCCP(1 uM) and Rotenone/Antimycin (1uM) at the time points specified.

**Mitochondrial morphology analysis.** MEFs were seeded in SensoPlate™ 24-Well Glass-Bottom Plate at the density of 50,000 cells per well and cultured for 4 h. The cells were treated with different concentrations of acetyl-l-carnitine for 20 h. MitoTracker® Green FM (Invitrogen, M7514) was added to cultural medium to make a final concentration of 50 nM MitoTracker® Green FM. Cells were incubated with MitoTracker® Green FM at 37 °C for 30 min; then the medium was removed and washed twice with PBS. Cells were analyzed with a Nikon Ti-E confocal microscope equipped with A1R scan head with spectral detector and resonant scanners using FITC filter. Images were quantified using Image J.

**Cathepsin B Activity assay.** MEFs were seeded in SensoPlate™ 24-Well Glass-Bottom Plate at the density of 50,000 cells per well and cultured for 24 h. Then cells were treated with 1X MagicRedTM (ImmunoChemistry Technologies, #937) staining solution for 30 min at 37 °C. Cells were analyzed with a Nikon Ti-E confocal microscope equipped with A1R scan head with spectral detector and resonant scanners using TRITC filter. Images were quantified using Image J.

**Histology and immunochemistry.** Animals were sacrificed and perfused with PBS followed by 4% paraformaldehyde. Brain sections (30 μM) were prepared on a cryostat. For immunostaining, tissue sections were mounted and blocked with 10% normal goat serum and 1% BSA, and then incubated with primary antibodies at 4 °C overnight. The primary antibodies are NEK1 (Abcam #ab229489, 1:200), NEK1 (Santa Cruz # sc-398813, 1:100), IBA-1 (Wako #019-19741, 1:1000), NeuN (EMD Millipore #MAB377X, 1:1000), chAT (EMD Millipore #AB144P, 1:500), CD31 (Invitrogen #MA1-40074, 1:500), α-synuclein (Invitrogen #AHB0261, 1:200), GR-1 (BioLegend #108401, 1:200), p-S166 RIPK1 (Biolynx #YJY-1-5, 1:200), p-S345 MLKL (CST #37333, 1:200), cleaved Caspase-3 (CST #9664, 1:200). Alexa Fluor secondary antibodies against the appropriate species were purchased from Life Technology and used according to the manufacturer's instructions.

**Thioflavin S staining.** Sections were rinsed in distilled water three times and placed in 0.5% HCl in 70% reagent grade alcohol for 5 s. Then sections were rinsed with distilled water and stained with 1% Thioflavin S solution for 5 min. Sections were washed and then placed in 1% acetic acid for 15 min. Sections were mounted to a glass slide for microscopy.

**BBB permeability assay.** P21 mice of indicated genotypes were intravenously injected with 10kd-tetramethyrhodamine (TMR)-labeled Dextran (Invitrogen, #D1868, 10 mg/mL) at the dose of 10 μl Dextran/g (bodyweight). Mice were sacrificed 20 min after injection and brains were fixed by immersion in 4% PFA at 4 °C overnight. Cryo-sections of cerebral cortex and thalamus at 30 μm thickness were prepared after the fixation and imaged by confocal microscopy.

**Protein expression and purification.** 293 T cells at 50% confluency in 150 mm dishes were transfected with expression vectors for HA tagged NEK1, FLAG tagged VPS26B$^{WT}$, VPS26B$^{S302/304A}$ for 48 hours (20 μg plasmid + 60 μl PEI per dish). Then cells were lysed with NP40 buffer containing 50 mM Tris (pH 7.4), 0.5% NP-40, 150 mM NaCl, 0.27 M sucrose, 2 mM EDTA, 2 mM EGTA, 50 mM NaF, 10 mM beta-glycerophosphate, 5 mM Na-pyrophosphate, 5 mM Na-orthovanadate, 0.1% BME, 1 mM PMSF, 2X Complete protease inhibitor cocktail freshly prepared. The cell lysates were immunoprecipitated with anti-FLAG or anti-HA antibody conjugated agarose at 4 °C overnight. Then the agarose was washed with high-salt lysis buffer [5% glycerol, 0.5% NP-40, 500 mM NaCl and 50 mM Tris–HCl (pH7.4)] for 5 times and then with water for 3 additional times. Flag-tagged

proteins and HA-tagged proteins were eluted with 0.1 M glycine pH 3.5 for 10 min at 25 °C and neutralized with 100 mM HEPES pH 7.4.

**In vitro kinase assay.** Purified FLAG-VPS26B and FLAG-VPS26B$^{S302A/S304A}$ was mixed with purified HA-NEK1 in a kinase reaction buffer (25 mM HEPES pH 7.4, 50 mM KCl, 10 mM MgCl2, 2X Roche's EDTA-free protease inhibitor cocktail, 0.1% BME) and ATP was added to 200 μM final concentration. Reaction was performed at 30 °C for 30 min with shaking at 1200 rpm and quenched by adding 5X SDS-PAGE sample buffer and heating at 95 °C for 5 min.

**Mass spectrometry and data analysis.** Cell surface proteins of $Nek1^{+/+}$ and $Nek1^{Kat2J/Kat2J}$ MEFs were purified by using Sulfo-NHS-SS-Biotin and NeutrAvidin Agarose and then precipitated with TCA. The precipitated proteins were subjected to trypsin digestion and LC-MS/MS analysis performed by Taplin Mass Spectrometry Facility, Harvard Medical School. LTQ Orbitrap Velos Pro ion-trap mass spectrometer (Thermo Fisher Scientific, Waltham, MA) and Sequest (Thermo Fisher Scientific, Waltham, MA) software program were used. To map phosphorylation sites in VPS26B by NEK1, we transfected 293 T cells with expression vectors for FLAG-VPS26B and equal amount of HA-NEK1$^{WT}$, or HA-NEK1$^{D146N/D146N}$ or HA for 24 h. Overexpressed protein was immobilized on anti-FLAG agarose beads (Sigma, M2) and followed by trypsin digestion on beads. The resulting peptides were subjected to the enrichment of phosphorylated peptides by using TiO2. The enriched phosphorylated peptides were analyzed on the Q Exactive HF-X mass spectrometer (Thermo Scientific). The identification and quantification of phosphorylated peptides was done by MaxQuant.

**Fluorescence microscopy.** $Nek1^{+/+}$, $Nek1^{Kat2J/Kat2J}$ MEFs and HeLa cells were fixed with 4% PFA at room temperature for 10 min, then blocked with 10% normal goat serum and 1% BSA, and then incubated with primary antibodies at 4 °C overnight. The primary antibodies are LAMP1(SantaCruz, #sc-19992), GLUT1 (Abcam, #ab40084), FLAG (Novus, #NB600-344). DyLight™ 650 Phalloidin was used to label F-actin (CST, #12956).

**Statistical analysis.** All cell death data and quantification of fluorescence were represented as mean ± s.d. of at least 3 replicates. Quantification of immunoblots and immunostaining were performed using ImageJ and CellProfiler. Data from plate reader were collected and analyzed with Graph Pad Prism 8 and Microsoft Excel 2016. All western blots from cell samples and mice tissues were repeated at least three times with similar results. A two-tailed Student's $t$ test was used for pairwise comparison between two groups. For multiple comparisons within three or more conditions with a single control, we performed one-way ANOVA followed by Dunnet's multiple comparison tests. Differences were considered to be significant if $p$-value < 0.05 (*). *$P$ < 0.05; **$P$ < 0.01; ***$P$ < 0.001; ns, no statistical significance.

## Data availability

The mass spectrometry proteomics data generated in this study have been deposited to the ProteomeXchange Consortium via the PRIDE[75] partner repository with the dataset identifier PXD027493 . All data are available in Source Data File. Uncropped western blot scans are shown in the Source Data File. Source data are provided with this paper.

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

## Acknowledgements

The authors thank Dr. Vishva Dixit of Genentech for K63 ubiquitin chain ab, Drs. Michelle Kelliher (University of Massachusetts) and Manolis Pasparakis (University of Cologne, Germany) for providing *Ripk1*$^{D138N}$ mice. We thank the Nikon Imaging Center at Harvard Medical School for assistance with microscopy. This work was supported in part by the HMS loyalty income (to JY), the China National Natural Science Foundation (21837004 and 91849204),the Chinese Academy of Sciences (XDB39030200) the National Key R&D Program of China (2016YFA0501900) and China National Natural Science Youth Foundation (31701210, 31801163).

## Author contributions

This project was conceived and directed by J.Y. The experiments were designed by J.Y. and H.W. H.W. conducted majority of the experiments. The manuscript was written by J.Y. and H.W. and edited by L.M. and Z.C. Z.Z. and Y.L. designed metabolic experiments. B.S. directed mass spectrometry experiments and analyzed mass spectrometry data. W. Q., C.Z., Z.X., M.Z., M.N., Z.L., L.M., A.N., and H.P. conducted some experiments.

## Competing interests

J.Y. is a consultant for Denali Therapeutics and Sanofi. The remaining authors declare no competing interest.
