## [Peer Review File · Nature Communications]

REVIEWER COMMENTS

Reviewer #1 (Remarks to the Author):

The manuscript by Wang et al. describes a mechanism where NEK1 controls and regulates retromer-mediated endosomal trafficking by phosphorylating one component of the retromer complex, VSP26B, that prevents the binding of SNX27, a major regulator of endosome-to-plasma membrane recycling of proteins. In conditions of NEK1 deficiency, this retromer complex gets disturbed affecting normal cellular homeostasis and leading to metabolic dysfunction through altered glucose uptake. In brain, this induces RIPK1 activation via suppression of A20, promoting necroptosis of endothelial cells, and leading to blood-brain barrier leakage. This induces neuroinflammation and accumulation of misfolded proteins, as seen in neurodegenerative diseases.

My main concern with this study is that the authors propose a mechanism which to my opinion is not more than a hypothesis trying to link different datasets and observations. However, how these observations are linked together is often not clear to me and not based on direct experimental evidence. Ex. How does metabolic dysfunction (as a result of the retromer complex alterations leading to defects in GLUT1 expression at the plasma membrane affecting glucose uptake) lead to reduced A20 expression and RIPK1 activation in endothelial cells ?

Other issues:

- Nek1-Kat2J mice are used in this study to model NEK1 deficiency: do these mice develop ALS-like pathology? If yes, then the authors should demonstrate that they have a motoneuron phenotype which is rescued in the RIPK1-D138 background. If not, what is the evidence that this is a model for human ALS in patients which express NEK1 variants ?
- The endosomal retromer complex was recently shown to be regulated by OTULIN via direct binding to SNX27 (Stangl et al., Nat. Commun., 2019). May there be a role for OTULIN in the observations shown in this study ?
- The postnatal lethality in Nek1-Kat2J mice can be partially rescued when RIPK1 kinase activity is inhibited. Can the phenotype be rescued by co-deletion of caspase-8 and RIPK3 ?
- Fig. 1g : Evans blue injection and quantification of BBB leakage. The original data should also be shown (picture of brain or brain section after Evans blue injection), especially since the fluorescence microscopy in Fig. 1h is not convincing (the 'diffuse' staining pattern can simply be a blurry picture).
- Fig. 2 : how does NEK1 regulate A20 levels ? Can A20 be directly phosphorylated by NEK1 ?
- Fig. 3f-g : enhanced expression of the DAM gene CST7. What about other DAM genes (ApoE, Trem2, Tyrobp, Axl, Ctsl, Cd9, Csf1, Itgax, Clec7a, Lilrb4, Timp2, Ctsd, Ctsb) ? Enhanced expression of CST7 is not enough to claim the presence of a DAM phenotype in microglia. What about expression of homeostatic genes (Tmem119, P2ry12, Cx3cr1, Sall1, Gpr34, Fcrls, Rhob, Olfml3) which are normally downregulated in DAM ?
- Fig. 6 : are these defects/differences (increased LAMP1 and p62 expression, mitochondrial fragmentation, size of lysosomes, AcetylCoA levels) also rescued in the RIPK-D138N genetic background ? Is there a role for autophagy ?
- TNFalpha should be replaced by TNF (since TNFbeta has been renamed to LT).

Reviewer #2 (Remarks to the Author):

Wang H presented a convincing work on the regulation of NEK1 on retromer-mediated endosomal trafficking by phosphorylating VPS26B. They especially showed that the metabolic and proteomic defects of NEK1 deficiency may modify the blood-brain barrier by sensitizing cerebrovascular endothelial cells to RIPK1 activation by promoting lysosomal degradation of A20, considered as a modulator of RIPK1.

This work is clearly impressive, but is hard to follow as too many results are provided and some discussion and methods are mixed to results. The field is not novel but is a promising way. The link between the different experiments must be improved and a schema to briefly resume the steps of the study is necessary. Some key experiments are shown exactly with the same importance as less relevant results (ex figure 1 and figure 5). And the general methodology is really not clear (mice management, cell experiments,...). Major revisions are recommended before publication.

-Although many experiments are rigorous and well designed, the interpretation and perspectives in ALS are not obvious.

The authors highlight the role of the retromer in regulating homeostasis of the CNS and explain how retromer dysfunction could be linked with neurodegeneration. They concluded that they could inhibit RIPK1 kinase for the treatment of ALS, AD, and also PD.

They present an activation of RIPK1 by a metabolic alteration, that finally may promote neuroinflammation, disrupted cellular homeostasis, and accumulation of misfolded proteins.

*However, metabolic homeostasis is dependent on the crosstalk neurone-astrocyte. GLUT1 (55Kda ?45 KDa?) are ubiquitous but neurons mainly expressed GLUT3? How is regulated GLUT3? How is the transfer of lactate from astrocytes to neurons in this context? The proof of metabolic suffering in this case is not shown enough and nothing is convincing about the specific suffering of neurons or motoneurons

*some references about the role of RIPK1 in ALS are not cited :

Necroptosis drives motor neuron death in models of both sporadic and familial ALS.

Re DB, et al. *Neuron*. 2014; Tuning Apoptosis and Neuroinflammation: TBK1 Restrains RIPK1. Yu H et al., *Cell*. 2018, Necroptosis is dispensable for motor neuron degeneration in a mouse model of ALS.

Wang T et al., 2020 , Axonal Degeneration: RIPK1 Multitasking in ALS. Politi K et al., *Curr Biol*. 2016; Targeting RIPK1 for the treatment of human diseases. Degterev A, Ofengeim D, Yuan J. *Proc Natl Acad Sci U S A*. 2019

*the main protein aggregations in ALS is TDP-43 aggregations that is not found or researched in this study. As it is largely considered as the hallmark of ALS, it is difficult to accept the interest of these findings in ALS according to the arguments provided in this manuscript

*as the weight loss is a major prognostic factor in ALS, how to interpret that inhibition of RIPK1 did not rescue the reduced body weight? And how to interpret in the context of metabolic disturbance and hypermetabolism in ALS for example?

*Ketogenic diets have been tested in ALS mice, patients but unfortunately this therapeutic strategy is not validated yet. It must be discussed, in regard to the perspectives of treatment suggested by the authors.

*About the neuroinflammation, the paper of Tortelli R, et al., (2020) about cytokines must be discussed

in regard to the results of the authors.

-other points

-What are all the ways of regulation previously described for NEK1? Are they the same as the other serine/threonine kinases? Are these ways different from the regulation of insulin receptor for example? Are AMPK, PI3K involved in these mechanisms?

-As A20 is a ubiquitin editing enzyme, it would be very interesting to discuss the ubiquitination modification in ALS and the impact of A20 modification

-How do the authors justify the choice of the use Evans blue?, rhodamine-labeled dextran? are the techniques used in routine practice in patients (albumin measurement in CSF and blood) not available? What are the interests of these techniques?

-Ketone bodies produced by the β -oxidation are preferentially used by mitochondrial respiration bypassing the glycolytic pathway and enhancing mitochondrial bioenergetics. The reduction of glycolysis may improve mitochondrial function, and may reduce apoptosis and inflammatory mediators. But, the mechanisms of action of ketogenic diets are diverse and their action is not only based on a metabolic effect. To note that the neuroprotective effect of specific polyunsaturated fatty acids and ketone bodies involved modulation of neuronal membrane excitability, inflammation, Reactive Oxygen Species production, or mitochondrial biogenesis,...it must be more discussed

-the choice of targeting GLUT1 among 79 substrates of the retromer is not clear.

-How can we explain (figure 5h) that G6P is higher than Glucose in the condition of Nek1 Kat2J/Kat2J? it is not the case for Nek1+/+; And the trend is also different for fructose 1-6 bisphosphate between both conditions. There is no difference for pyruvate. Have the authors explored the activity of PDH? PC? How can the authors explain this result? Have they measured lactate? The ratio lactate/pyruvate is always more relevant than pyruvate alone.

-Have the authors evaluated the role of p-AAC on fatty acid beta-oxidation? What is the link between p-AAC, malonylCoA, beta-oxidation? And glucose metabolism by the way.

-a discussion about acetylCoA must contain few words on the potential other origins of acetylCoA, and other potential actors of energetic metabolism (like aminoacids)

-The authors often started to study some important actors, necessary to explain the mechanism of NEK1 regulation but it is often not complete. So different ways are opened but we have no global overview of the metabolism disturbance.... It is difficult to suggest a final convincing mechanism with missing data (mitochondria study is not complete, energetic metabolism is not fully explored,...). So some conclusions may be overinterpreted

Minor points

-The introduction is too long and masks the workflow of this study.

-some statistics data are often missing in the results part and it could be helpful to provide some of them in the text, thus avoiding to systematically have a look on the figures.

-How could the authors explain the dispersion of the results in figure 3 d within the subgroup of Nek1 Kat2J/Kat2J? In the figure 3, some inflammation markers are measured but their specific role is not clear.

-seahorse findings are not explained enough in the context of other metabolic data

-Have the authors evaluated the role of pentose phosphate pathway in their experiments to see the proportion between glycolysis and PPP?

-What are the main criteria to highlight candidates in the volcano plot (fig 4)?

Reviewer #3 (Remarks to the Author):

The manuscript by Wang et al provides critical evidence for NEK1's functions in regulating retromer, blood brain barrier (BBB) integrity, glucose metabolism and RIPK1 activation. NEK1, a serine/threonine kinase, is a risk gene for ALS. This paper shows that NEK1 phosphorylates VPS26B, a component of retromer; and deficiency in NEK1 results in multiple phenotypes, including disrupted endosomal trafficking of plasma membrane proteins, dysfunctional mitochondria and lysosomes, accumulation of a-synuclein, RIPK1 activation, and impaired BBB. Inactivation of RIPK1 or feeding with ketogenic diet in NEK1 deficit mice can prevent their postnatal lethality and BBB damage. These results reveal a novel regulatory mechanism of retromer complex and its function, and uncover insights into NEK1 regulation of ALS pathogenesis. While this paper is of importance to the field, a few concerns remain to be addressed.

Major concerns:

First, this manuscript covers several proteins' functions in multiple aspects, including NEK1, VPS26b, RIPK1,... Thus, many gaps in between remain, and many questions raised. For example, it lacks of evidence to functionally link VPS26b with the NEK1's functions in regulating BBB and RIPK1 activation. Does VPS26b deficiency show similar phenotypes as that of NEK1 mutant mice? Does expression of VPS26b or alter VPS26b's phosphorylation diminish the phenotypes in NEK1 mutant mice? How does NEK1 regulate RIPK1 activation?

Second, this study showed that *Nek1Kat2j/Kat2j* reduced the phosphorylation of VPS26B (Fig 4e); the non-phosphomimetic VPS26B (S302A/S304A) increased its binding with SNX27 (a retromer complex that promotes surface levels of GLUT1)(...), but decreased its binding with SNX2 (Fig 4g); and the binding of phosphomimetic VPS26B (S302D/S304D) with SNX27 was reduced. However, *Nek1Kat2j/Kat2j* also showed significantly reduced plasma membrane distribution of GLUT1, while the SNX27 and VPS26B interaction was increased. How to explain these results? It would be better to illustrate clearly the exactly working model or hypothesis.

Other concerns:

1. There are two panels 'e' in the extended Fig.1, the second one should be 'f'.
2. Fig 1b only shows the survival rate of 3 groups of mice, the 3 missing groups should be included.
3. The molecular weight of each band should be marked on the western blot images. For example, some proteins in Fig 2 have multiple bands, which is confusing.
4. The description of Fig 2i is unclear. The manuscript (Page 9) describes that 'a reduction of A20 protein in *Nek1Kat2j/Kat2j* compared to WT mice', while the figure shows '*Nek1+/+; RIPK1D138N/D138N*' compared to '*Nek1Kat2j/Kat2j; RIPK1D138N/D138N*'.
5. Fig 3a only shows a small area of thioflavin S staining, and it is difficult to reflect the change in the number of thioflavin S+ cells. Could the author show a larger brain area to include more positive cells?
6. In Fig 3c, compared with *Nek1+/+* group, *Nek1Kat2j/Kat2j; RIPK1D138N/D138N*' showed obviously more a-Synuclein positive cells, however, this appears to be in-consistent with the quantification data in Fig 3d.

Reviewer #1 (Remarks to the Author):

The manuscript by Wang et al. describes a mechanism where NEK1 controls and regulates retromer-mediated endosomal trafficking by phosphorylating one component of the retromer complex, VSP26B, that prevents the binding of SNX27, a major regulator of endosome-to-plasma membrane recycling of proteins. In conditions of NEK1 deficiency, this retromer complex gets disturbed affecting normal cellular homeostasis and leading to metabolic dysfunction through altered glucose uptake. In brain, this induces RIPK1 activation via suppression of A20, promoting necroptosis of endothelial cells, and leading to blood-brain barrier leakage. This induces neuroinflammation and accumulation of misfolded proteins, as seen in neurodegenerative diseases.

My main concern with this study is that the authors propose a mechanism which to my opinion is not more than a hypothesis trying to link different datasets and observations. However, how these observations are linked together is often not clear to me and not based on direct experimental evidence. Ex. How does metabolic dysfunction (as a result of the retromer complex alterations leading to defects in GLUT1 expression at the plasma membrane affecting glucose uptake) lead to reduced A20 expression and RIPK1 activation in endothelial cells ?

Reply:

We conducted following experiments in order to further connect the mechanism as requested by this reviewer.

1) To address this question how metabolic dysfunction in Nek1 deficient cells may reduce the levels of A20, we investigated the effect of metabolism on A20. Our new data shows that the metabolic defects promote necroptosis by reducing the acetylation of A20 (shown in the following figure which is also presented as Extended Data Fig 6i), which in turn promotes lysosomal degradation of A20. Either blocking the lysosomal degradative pathway or promoting fatty acid metabolism by acetyl-L-carnitine and octanoate rescued the decreased A20 protein level in Nek1^{Kat2J/Kat2J} cells (Fig. 2e, 6i; Extended Data Fig. 6j). This is consistent with what we have recently reported that reduced acetylation of A20 in endothelial cells promotes its degradation through the lysosome pathway, which is involved in mediating BBB damage in Alzheimer's disease (Zou et al., 2020).

Extended Data Fig. 6i, Decreased lysine acetylation of A20 in Nek1^{Kat2J/Kat2J} MEFs. Total lysates of Nek1^{+/+} and Nek1^{Kat2J/Kat2J} were normalized for equal amount of A20 protein. Then acetylated proteins were immunoprecipitated with anti-acetylated-lysine agarose and analyzed by western blotting.

2) To further demonstrate the functional significance and mechanism by which NEK1 regulates retromer function by phosphorylating VPS26B, we analyzed the effect of VPS26B^{S302A/S304A} and VPS26B^{S302D/S304D} on the trafficking of SNX27 and SNX2. Consistent with increased plasma membrane presence of SNX27 in Nek1^{Kat2J/Kat2J} MEFs compared to that of WT, the levels of SNX27 in the cell surface membrane were increased in Nek1^{Kat2J/Kat2J} MEFs expressing

VPS26B^{S302A/S304A} and reduced in cells expressing VPS26B^{S302D/S304D}. In contrast, the levels of SNX2 in the cell surface membrane were decreased in VPS26B knockdown MEFs expressing VPS26B^{S302A/S304A} and increased in VPS26B knockdown MEFs expressing VPS26B^{S302D/S304D} (Extended Data Fig. 4l). In addition, we further confirmed the increased and decreased plasma membrane levels of SNX27 and SNX2, respectively, in Nek1^{Kat2J/Kat2J} MEFs compared to that of WT by western blotting analysis after cell surface selective aminoxy-biotinylation (Fig. 5a). Together with the previous data, these results allow us to propose a model for the molecular mechanism that Nek1 deficiency blocks the phosphorylation of S302/S304 VPS26B which in turn leads to mislocalization of SNX27- and SNX2-retromers (Extended Data Fig. 7g). Phosphorylation of S302/S304 VPS26B by NEK1 modulates the recycling of SNX27-retromer and SNX2-retromer that deliver proteins from endosomes to cell surface membrane and lysosomes. Thus, NEK1 deficiency blocks the recycling of SNX27-retromer from cell surface to endosomes which inhibits further trafficking of SNX27-retromer from endosomes to cell surface. Since SNX27-retromer is important for delivering GLUT1 to the cell surface membrane, NEK1 deficiency leads to reduced GLUT1 levels at the cell surface membrane which reduces glucose uptake. Reduced glucose uptake disrupts mitochondrial metabolism and reduces cellular levels of acetyl-CoA. Reduced acetyl-CoA inhibits the acetylation of A20 which promotes its lysosomal degradation (Zou et al., 2020). Reduced levels of A20 sensitizes to the activation of RIPK1-dependent cell death and inflammation which can be inhibited by Nec-1s and also by ketogenic diet which provides alternate source of acetyl-CoA to restore cellular levels of A20.

We would like to summarize our overall experimental paradigm and the major results using this flow chat in the figures bellow (which is also presented as Extended Data Fig. 7f). The primary observations in this manuscript include the premature postnatal lethality, BBB damage and neuroinflammation of Nek1-Kat2J mice could be rescued by inhibition of RIPK1 and also by ketogenic diet. At cellular levels, Nek1 deficiency promotes metabolic defects and RIPK1-dependent cell death. Nek1 directly mediates the phosphorylation of VPS26B to regulate retromer function. Defective retromer function in Nek1 deficient cells reduces the levels of GLUT1 on membrane and uptake of glucose into the cells. Reduced glucose levels impair glycolysis and levels of acetyl-CoA, a key substrate for protein acetylation. Thus, the acetylation of A20 is reduced in Nek1 deficient cells which promotes the lysosomal degradation of A20. Reduced levels of A20 in different lineages have been shown to model RIPK1-mediated inflammatory diseases (Mifflin et al., 2020).

Extended Data Fig. 7f. Flow-chart of this study.

We agree with this reviewer that we employed a wide range of techniques and obtained many different datasets to uncover the mechanism by which Nek1 deficiency promotes cellular dysfunction and the activation of RIPK1. Importantly, our study uncovered a molecular mechanism that is highly novel and supported by extensive amount of experimental evidence, including:

- 1) NEK1 phosphorylates VPS26B to regulate retromer trafficking;
- 2) NEK1 deficiency promotes mis-localization of membrane receptors;
- 3) NEK1 deficiency leads to mis-localization of GLUT1 and defects in glucose-uptake, which in turn leads to defects in mitochondrial metabolism and reduced levels of acetyl-CoA, a key substrate of protein acetylation and reduced levels of acetylated A20 and increased lysosomal degradation of A20;
- 4) NEK1 deficiency leads to unexpected defects in BBB damage which in turn promotes neuroinflammation and α -synuclein aggregation. Although α -synuclein aggregation and neuroinflammation are common in human neurodegenerative diseases, no genetic loss-of-function mutation in mice has been reported to lead to this phenotype, which underlies the potential relevance and importance of our study to human neurodegenerative diseases beyond what is described in this manuscript;
- 5) NEK1 deficiency promotes the aggregation of α -synuclein, a hallmark of PD, and the role of RIPK1 in promoting the aggregation of α -synuclein and thus, provides unexpected connections from metabolic defects and neuroinflammation with the aggregation of α -synuclein.

Other issues:

- Nek1-Kat2J mice are used in this study to model NEK1 deficiency: do these mice develop ALS-like pathology? If yes, then the authors should demonstrate that they have a motoneuron phenotype which is rescued in the RIPK1-D138 background. If not, what is the evidence that this is a model for human ALS in patients which express NEK1 variants?

Reply: Nek1-Kat2J mice die young (before 3 weeks of age) due to BBB damage and other prenatal developmental defects before their neurological defects can be functionally assessed. The goal of this 5-year+ study is to investigate the null phenotype of Nek1 in mice. Homozygous LoF (loss-

of-function) *NEK1* variants are a cause of autosomal-recessive lethal osteochondrodysplasia, a severe early developmental disorder known as short rib polydactyly syndrome type II (SRPS)(Thiel et al., 2011), and Jeune asphyxiating thoracic dystrophy(McInerney-Leo et al., 2015); while mutant *NEK1* alleles associated with ALS are reduction-of-function in nature. *NEK1* has been implicated in mediating diverse cellular functions, including cell cycle, cilia formation, DNA-damage response, microtubule stability, and neural development (Fry et al., 2012). It remains unclear how *NEK1* may mediate these diverse functions at cellular levels; nor is it known why *NEK1* deficiency in animals and humans can lead to early developmental bone disorders as well as sensitize to developing ALS later in life. Our study uncovered the role of *NEK1* as an important regulator of retromer which regulates protein trafficking to the cell membrane. Our discovery nicely explains why *Nek1* deficiency may show such pleiotropic defects because the impact of retromer on many aspects of cellular physiology. The goal of this study is not to provide another mouse model of ALS, as *Nek1* mutations found in ALS are mostly point mutations and heterozygous. Our study provides the foundation for future studies to further investigate the mechanism of *Nek1* in ALS.

- The endosomal retromer complex was recently shown to be regulated by OTULIN via direct binding to SNX27 (Stangl et al., Nat. Commun., 2019). May there be a role for OTULIN in the observations shown in this study?

Reply: To address this question, we investigated the effect of *Nek1* deficiency on OTULIN levels. We found that the protein level of OTULIN is not affected by *NEK1* deficiency as shown in the figure below.

Legend: Total lysis of *Nek1*^{+/+} and *Nek1*^{Kat2J/Kat2J} MEFs were analyzed by immunoblotting with indicating antibodies.

- The postnatal lethality in *Nek1-Kat2J* mice can be partially rescued when *RIPK1* kinase activity is inhibited. Can the phenotype be rescued by co-deletion of caspase-8 and *RIPK3*?

Reply: *Nek1-Kat2J* mice suffer from defects due to reduced uptake of glucose, defects in mitochondrial metabolism and lysosomes from dysfunctional retromer-mediated transport, which cannot be inhibited by blocking *RIPK1* kinase and is the likely explanation for the remaining lethality in this mutant line. Since co-deletion of caspase-8 and *RIPK3* is not known to affect metabolism, mitochondria or lysosome, it is unlikely to provide further protection than inhibition of *RIPK1*. This is also consistent with the pleiotropic defects as the result of retromer dysfunction which affects the cell surface distribution of many receptors, which cannot be directly inhibited by *RIPK1* inhibition or unlikely by co-deletion of caspase-8 and *RIPK3*.

- Fig. 1g : Evans blue injection and quantification of BBB leakage. The original data should also be shown (picture of brain or brain section after Evans blue injection), especially since the fluorescence microscopy in Fig. 1h is not convincing (the 'diffuse' staining pattern can simply be a blurry picture).

Reply: The quantification of Evans blue levels in brains was measured by using a standard protocol in the field with a plate reader in total brain lysate of the mice intraperitoneally injected with Evans blue and after PBS perfusion; thus, the measurement is quantitative and unbiased (Fig. 1g). In addition, we provide new fluorescence images of rhodamine-dextran staining with higher resolution that shows the image is in focus (Fig.1h). We apologize for the poor quality of the original figure which was made blurry during the conversion to pdf.

Legend: **h**, Mice (3 weeks old) were intravenously injected with 10kD-tetramethylrhodamine (TMR)-labeled Dextran and then sacrificed. The cryo-sections of cerebral cortex were imaged by confocal microscope.

- Fig. 2 : how does NEK1 regulate A20 levels ? Can A20 be directly phosphorylated by NEK1 ?

Reply: In this revised manuscript, we provide new data to show that metabolic defects in $Nek1^{Kat2J/Kat2J}$ cells reduced acetyl-CoA level (Extended Data Fig. 5e) and thus reduced acetylation of A20 protein (Extended Data Fig. 6i), which in turn promotes lysosomal degradation of A20. Reduced acetylation of A20 is known to promote its lysosomal degradation (Zou et al., 2020). Since the upshifted A20 phosphorylation band was not affected by $Nek1$ deficiency, it is unlikely that $Nek1$ can directly phosphorylate A20.

- Fig. 3f-g : enhanced expression of the DAM gene CST7. What about other DAM genes (ApoE, Trem2, Tyrobp, Axl, Ctsl, Cd9, Csf1, Itgax, Clec7a, Lilrb4, Timp2, Ctsd, Ctsb) ? Enhanced expression of CST7 is not enough to claim the presence of a DAM phenotype in microglia. What about expression of homeostatic genes (Tmem119, P2ry12, Cx3cr1, Sall1, Gpr34, Fcrls, Rhob, Olfml3) which are normally downregulated in DAM ?

Reply: We thank this reviewer for asking this insightful question. *Cst7* is transcriptionally regulated by RIPK1 kinase activity (Ofengeim et al., 2017). To answer the reviewer's question about DAM genes and homeostatic genes, we investigated the levels of DAM by qPCR (shown in Extended Data Fig. 3d-e). Following the advice of this reviewer, we examined the levels of all DAM markers defined by Amit group (Keren-Shaul et al., 2017). Consistent with the induction of DAM, in addition to *Cst7*, the expression of all other known DAM signature genes including *ApoE*, *Trem2* (6.14x), *Tyrobp* (4.26x), *Axl*, *Ctsl*, *Cd9*, *Csf1*, *Clec7a* (77.17x), *Lilrb4* (44.81x), *Ctsd*, *Ctsb* and *Itgax* (10.79x), except *Timp2*, were also highly elevated in the brains of $Nek1^{Kat2J/Kat2J}$ mice, which were suppressed by the inhibition of RIPK1 in $Nek1^{Kat2J/Kat2J};RIPK1^{D138N/D138N}$ mice (Extended Data Fig. 3d). In particular, some of these RIPK1-regulated DAM markers, such as *ApoE*³⁴, *Trem2*³⁵, *Tyrobp*³⁶ and *Ctsd*³⁷, are known AD risk factors. In contrast, the markers of homeostatic microglia, such as *Tmem119*, *R2ry12*, *Cx3cr1*, *Sall1*, and *Rhob*²⁹, were not elevated, or minimally induced, such as *Gpr34* (1.85x), *Fcrls* (3.61x) and *Olfml3* (3.04x), in the brains of $Nek1^{Kat2J/Kat2J}$ mice, which were also suppressed by the inhibition of RIPK1 in the brains of $Nek1^{Kat2J/Kat2J};RIPK1^{D138N/D138N}$ mice (Extended Data Fig. 3e).

- Fig. 6 : are these defects/differences (increased LAMP1 and p62 expression, mitochondrial fragmentation, size of lysosomes, AcetylCoA levels) also rescued in the RIPK-D138N genetic background ? Is there a role for autophagy ?

Reply: RIPK1-D138N is a well-established kinase dead mutation of RIPK1. To answer this reviewer question if inhibition of RIPK1 might have any effect on lysosome, autophagy and mitochondria, we provide new data of Nek1^{Kat2J/Kat2J} MEFs treated with Nec-1s, a well-established RIPK1 inhibitor, on Lamp1, p62, and LC3 blots, lysotracker and mitotracker staining and acetyl-CoA levels (Fig. 6f, Extended Data Fig. 6k-n). Inhibition of RIPK1 by Nec-1s has no effects on the increased LAMP1 and p62 levels, mitochondrial fragmentation, size of lysosomes and decreased acetyl-CoA levels. LC3II level is not affected by NEK1 deficiency. These new data are presented as Fig 6f. Thus, Nek1 deficiency does not directly impact on autophagy induction.

- TNFalpha should be replaced by TNF (since TNFbeta has been renamed to LT).

Reply: We replaced every TNFalpha with TNF in the manuscript.

Reviewer #2 (Remarks to the Author):

Wang H presented a convincing work on the regulation of NEK1 on retromer-mediated endosomal trafficking by phosphorylating VPS26B. They especially showed that the metabolic and proteomic defects of NEK1 deficiency may modify the blood-brain barrier by sensitizing cerebrovascular endothelial cells to RIPK1 activation by promoting lysosomal degradation of A20, considered as a modulator of RIPK1.

This work is clearly impressive, but is hard to follow as too many results are provided and some discussion and methods are mixed to results. The field is not novel but is a promising way. The link between the different experiments must be improved and a schema to briefly resume the steps of the study is necessary. Some key experiments are shown exactly with the same importance as less relevant results (ex figure 1 and figure 5). And the general methodology is really not clear (mice management, cell experiments,...). Major revisions are recommended before publication.

Reply: We thank this reviewer for the strong support. We have conducted new experiments in order to address all of the questions from this reviewer. We have also worked hard to improve the flow of the manuscript and clarification. To increase the readability of this manuscript, we added a summary flow chart of this study and the graphic model of the molecular mechanism (Extended Data Fig. 7f-g).

-Although many experiments are rigorous and well designed, the interpretation and perspectives in ALS are not obvious.

The authors highlight the role of the retromer in regulating homeostasis of the CNS and explain how retromer dysfunction could be linked with neurodegeneration. They concluded that they could inhibit RIPK1 kinase for the treatment of ALS, AD, and also PD.

They present an activation of RIPK1 by a metabolic alteration, that finally may promote neuroinflammation, disrupted cellular homeostasis, and accumulation of misfolded proteins.

Reply: Nek1-Kat2J mice die early (before 3 weeks of age) due to BBB damage and other prenatal developmental defects before their neurological defects can be functionally assessed. The goal of this study is to investigate the null phenotype of Nek1 in mice. Homozygous LoF (loss-of-function) *NEK1* variants are a cause of autosomal-recessive lethal osteochondrodysplasia, a severe early developmental disorder known as short rib polydactyly syndrome type II (SRPS)(Thiel et al., 2011), and Jeune asphyxiating thoracic dystrophy(McInerney-Leo et al., 2015); while mutant NEK1 alleles associated with ALS are reduction-of-function in nature. NEK1 has been implicated in mediating diverse cellular functions, including cell cycle, cilia formation, DNA-damage response, microtubule stability, and neural development (Fry et al., 2012). It remains unclear how NEK1 may mediate these diverse functions at cellular levels; nor is it known why NEK1 deficiency in animals and humans can lead to early developmental bone disorders as well as sensitize to developing ALS later in life. Our study uncovered the role of NEK1 as an important regulator of retromer which regulates protein trafficking to the cell membrane. Our discovery nicely explains why Nek1 deficiency may show such pleiotropic defects because the impact of retromer on many aspects of cellular physiology. The goal of this 5-year+ study is not to provide another mouse model of ALS, as Nek1 mutations found in ALS are mostly point mutations and heterozygous. Our study provides the foundation for future studies to further investigate the mechanism of Nek1 in ALS.

The discovery described in this manuscript demonstrated the role of RIPK1 in mediating cell death and inflammation caused by Nek1 deficiency and possibility of inhibiting RIPK1 for the treatment

of neurodegenerative diseases in ALS, AD and PD as retromer defects have been implicated in these diseases; but importantly, this study also showed the limitation of inhibiting RIPK1, as inhibition of RIPK1 cannot rescue the lysosomal and mitochondrial defects caused by Nek1 deficiency, as inhibition of RIPK1 by Nec-1s has no effects on the increased LAMP1 and p62 expression, mitochondrial fragmentation, size of lysosomes and decreased glucose uptake and acetyl-CoA levels in Nek1^{Kat2J/Kat2J} cells (Fig. 6f, Extended Data Fig. 6k-n). Since retromer defects lead to metabolic dysfunction, this study explored and demonstrated an alternative way to ameliorate neurodegeneration by ketogenic diet.

*However, metabolic homeostasis is dependent on the crosstalk neurone-astrocyte. GLUT1 (55Kda ?45 KDa?) are ubiquitous but neurons mainly expressed GLUT3? How is regulated GLUT3? How is the transfer of lactate from astrocytes to neurons in this context? The proof of metabolic suffering in this case is not shown enough and nothing is convincing about the specific suffering of neurons or motoneurons

Reply: The molecular weight of GLUT1 shown in Fig.5a is ~55KDa, we added the molecular weight of related protein markers for all western blot results in the figures. To address this question, we investigated the protein level of GLUT3 and its distribution on cell plasma membrane in Nek1^{Kat2J/Kat2J} MEFs. As shown in the figure below, there is no difference of GLUT3 level in either protein level in total lysis or its distribution on plasma membrane in Nek1^{Kat2J/Kat2J} MEFs compared to Nek1^{+/+} MEFs.

Since Nek1-Kat2J mice die before birth or soon after birth, we focused on investigating the reason why these mice die so early, which led to the discovery of dramatic BBB damage in these newborn Nek1^{Kat2J/Kat2J} mice, which can be rescued by inhibition of RIPK1 and ketogenic diet. Since defects in metabolism of astrocytes or neurons are unlikely to explain the early lethality, we did not investigate the metabolism of neuron and astrocyte. Since our study suggests that Nek1 deficiency promotes metabolic defects, it is definitely worth in future study to investigate mechanism in mice with Nek1 point mutations that mimic ALS mutations in humans.

Legend: Total lysis and plasma membrane fraction by biotinylated cell surface membrane pull-down of Nek1^{+/+} and Nek1^{Kat2J/Kat2J} MEFs were analyzed by indicating antibodies.

*some references about the role of RIPK1 in ALS are note cited : Necroptosis drives motor neuron death in models of both sporadic and familial ALS. Re DB, et al S. Neuron. 2014; Tuning Apoptosis and Neuroinflammation: TBK1 Restrains RIPK1. Yu H et al., Cell. 2018, Necroptosis is dispensable for motor neuron degeneration in a mouse model of ALS. Wang T et al., 2020 , Axonal Degeneration: RIPK1 Multitasking in ALS. Politi K et al., Curr Biol. 2016; Targeting RIPK1 for the treatment of human diseases. Degterev A, Ofengeim D, Yuan J. Proc Natl Acad Sci U S A. 2019

Reply: These citations have been added.

*the main protein aggregations in ALS is TDP-43 aggregations that is not found or researched in this study. Ad it is largely considered as the hallmark of ALS, it is difficult to accept the interest of these findings in ALS according to the arguments provided in this manuscript

Reply: Since $Nek1^{Kat2J/Kat2J}$ (a total loss-of-function *Nek1* mutation) mice die early (before 3 weeks of age), it is not a good model for ALS, which is associated with reduction-of-function mutations in *Nek1*, such as p.Arg261His mutation (Kenna et al., 2016). Homozygous LoF *NEK1* variants are a cause of autosomal-recessive lethal osteochondrodysplasia, a severe early developmental disorder known as short rib polydactyly syndrome type II (SRPS) (Thiel et al., 2011). *NEK1* has been implicated in mediating diverse cellular functions, including cell cycle, cilia formation, DNA-damage response, microtubule stability, and neural development (Fry et al., 2012). This study is to begin to address how *NEK1* may mediate these diverse functions at cellular levels. As discussed before, since $Nek1^{Kat2J/Kat2J}$ mice die before birth or soon after birth, we focused on investigating the reason why these mice die so early, which led to the discovery of dramatic BBB damage in these newborn $Nek1^{Kat2J/Kat2J}$ mice. The premature lethality of this mutant line also precludes the analysis of motor dysfunction in later life. Future studies are needed to investigate the phenotype of point mutations in *NEK1* that mimic what is found in humans ALS.

*as the weight loss is a major prognostic factor in ALS, how to interpret that inhibition of RIPK1 did not rescue the reduced body weight? And how to interpret in the context of metabolic disturbance and hypermetabolism in ALS for example?

Reply: The weight loss in *Nek1-Kat2J* mice is likely due to dysfunctional retromer which affects multitude of cellular physiology. It is unclear if metabolic disturbance and hypermetabolism in ALS is related to RIPK1. As discussed before, our study is consistent with the specific role of RIPK1 in mediating cell death and neuroinflammation and thus, if the metabolic disturbance in human ALS is not directly related to neuroinflammation, inhibition of RIPK1 may not improve the metabolic disturbance in ALS patients.

*Ketogenic diets have been tested in ALS mice, patients but unfortunately this therapeutic strategy is not validated yet. It must be discussed, in regard to the perspectives of treatment suggested by the authors.

Reply: We provide further discussion on this point.

*About the neuroinflammation, the paper of Tortelli R,et al., (2020) about cytokines must be discussed in regard to the results of the authors.

Reply: This paper has been cited.

-other points

-What are all the ways of regulation previously described for *NEK1*? Are they the same as the other serine/threonine kinases? Are these ways different from the regulation of insulin receptor for example? Are AMPK, PI3K involved in these mechanisms?

Reply: HIF-2 α has been suggested to mediate hypoxia induced *NEK1* upregulation (Chen et al., 2019b). The Tausled Like kinases (TLKs) have been shown to bind and regulate *NEK1* (Singh et al., 2017). But there is no report that AKT or PI3K regulate *NEK1*.

-As A20 is a ubiquitin editing enzyme, it would be very interesting to discuss the ubiquitination modification in ALS and the impact of A20 modification

Reply: A section is added in Discussion on the ubiquitination modification in ALS and the impact of A20 modification. Reduced A20 levels have been found in the whole blood of patients with MS and PD (Perga et al., 2017).

-How do the authors justify the choice of the use Evans blue?, rhodamine-labeled dextran?are the techniques used in routine practice in patients (albumin measurement in CSF and blood) not available? What are the interests of these techniques?

Reply: Evans blue and rhodamine-labeled dextran are two well-established methods for measuring BBB integrity in animal studies (Andrade et al., 2018; Chen et al., 2019a; Choi et al., 2011; Daneman et al., 2010; Haruwaka et al., 2019; Jiang et al., 2020; Lin et al., 2012; Minutti et al., 2019; Reyahi et al., 2015; Tager et al., 2008).

- Ketone bodies produced by the β -oxidation are preferentially used by mitochondrial respiration bypassing the glycolytic pathway and enhancing mitochondrial bioenergetics. The reduction of glycolysis may improve mitochondrial function, and may reduce apoptosis and inflammatory mediators. But, the mechanisms of action of ketogenic diets are diverse and their action is not only based on a metabolic effect To note that the neuroprotective effect of specific polyunsaturated fatty acids and ketone bodies involved modulation of neuronal membrane excitability, inflammation, Reactive Oxygen Species production, or mitochondrial biogenesis,...it must be more discussed

Reply: We have added a section in Discussion about diverse effect of ketogenic diets.

-the choice of targeting GLUT1 among 79 substrates of the retromer is not clear.

Reply: We chose GLUT1 is to explain the defects in glucose uptake and metabolism that we found: because 1) GLUT1 is a well-established substrate of retromer; 2) Nek1 deficient cells have a defect in glucose uptake; 3) GLUT1 is known to play a major role in glucose uptake. Our study does not rule out the role of other GLUT family members which may also mediate glucose-uptake in cell type-specific manner.

-How can we explain (figure 5h) that G6P is higher than Glucose in the condition of Nek1 Kat2J/Kat2J? it is not the case for Nek1^{+/+}; And the trend is also different for fructose 1-6 bisphosphate between both conditions. There is no difference for pyruvate. Have the authors explored the activity of PDH? PC? How can the authors explain this result? Have they measured lactate? The ratio lactate/pyruvate is always more relevant than pyruvate alone.

Reply: We thank this reviewer for asking this question. In the original figure 5h, the intensities of metabolites were expressed as peak areas. In LC-MS analysis, due to different ionization efficiencies between metabolites, the peak areas could only be used to compare the same metabolite between different groups, e.g. between Nek1^{+/+} and Nek1^{Kat2J/Kat2J} groups. Thus, the peak area could not be used to compare between different metabolites, such as G6P and Glucose. We apologize for the confusion caused by this misleading figure. In the revised manuscript, we have modified the Figure 5h by normalizing each metabolite intensities in Nek1^{+/+} group as 1. After data normalization, the original conclusion stays the same.

As to why no difference in pyruvate levels in Figure 5h: this was a fast (7 minutes) ^{13}C labeling experiment. Since pyruvate is a downstream product of glucose, 7 minutes might not be enough to allow sufficient amount of labeled pyruvate to be generated. To test this hypothesis, we conducted a longer term ^{13}C labeling experiment (2 hours). Consistent with our hypothesis, the levels of pyruvate_M+3 showed a significant decrease in Nek1^{Kat2J/Kat2J} MEFs ($P < 0.05$). This data is now presented as Extended Data Fig. 5c.

In metabolomics experiment without ^{13}C labelling, the ratio of lactate/pyruvate is more relevant than pyruvate alone since pyruvate could be generated from multiple different pathways. Since in our fast ^{13}C -glucose labeling experiment, pyruvate_M+3 was all generated from isotope labelled ^{13}C -glucose, which could directly reflect the activity of glycolysis. Thus, the data from pyruvate_M+3 is enough to characterize the glycolysis activity. PDH (Pyruvate dehydrogenase) and PC (Pyruvate carboxylase) are not cell-surface receptors such as GLUT1. Since the reviewer asked for PDH and PC, we analyzed phosphorylation of PDH and protein level of PDH and PC. There is no difference of them in between Nek1^{+/+} and Nek1^{Kat2J/Kat2J} MEFs as shown in the figure below.

Legend: Total lysis of Nek1^{+/+} and Nek1^{Kat2J/Kat2J} MEFs were analyzed by immunoblotting with indicating antibodies.

-Have the authors evaluated the role of p-AAC on fatty acid betaoxydation? What is the link between p-ACC, malonylCoA, betaoxydation? And glucose metabolism by the way.

Reply: To answer this question, we analyzed the level of malonyl-CoA and the ratio of Acetyl-carnitine to carnitine in Nek1^{+/+} and Nek1^{Kat2J/Kat2J} MEFs by mass spectrometry. Both the level of malonyl-CoA and the ratio of acetyl-carnitine to carnitine showed no significant difference in Nek1^{Kat2J/Kat2J} MEFs compared to that of Nek1^{+/+} MEFs. These data suggest that p-ACC has no obvious effect on malonyl-CoA and fatty acid beta-oxidation in this situation as shown in the following figures.

Legend: Ratio of Acetyl-carnitine to carnitine in Nek1^{+/+} and Nek1^{Kat2J/Kat2J} MEFs determined by mass spectrometry (mean ± SD, n=6).

Legend: The level of malonyl-CoA in Nek1^{+/+} and Nek1^{Kat2J/Kat2J} MEFs determined by mass spectrometry (mean ± SD, n=6).

-a discussion about acetylCoA must contain few words on the potential other origins of acetylCoA, and other potential actors of energetic metabolism (like aminoacids)

Reply: We added the potential other origins into the discussion of the manuscript.

-The authors often started to study some important actors, necessary to explain the mechanism of NEK1 regulation but its is often not complete. So different ways are opened but we have no global overview of the metabolism disturbance.... It is difficult to suggest a final convincing mechanisms with missing data (mitochondria study is not complete, energetic metabolism is not fully explored,...). So some conclusions may be overinterpreted

Reply:

To get a global view of the metabolism disturbance, we compared fast ¹³C-glucose labeling and long-term (2 hours) ¹³C-glucose labeling assay by mass spectrometry. In addition to our data of fast ¹³C-glucose labeling data that shows decreased glucose uptake and glycolysis in Nek1^{Kat2J/Kat2J} MEFs, the levels of pyruvate_{M+3} in longer term ¹³C-glucose labeling experiment (2 hours) also showed a significant decrease in Nek1-Kat2J MEFs (P<0.05, data is presented in Extended Data Fig. 5c) which also supports the idea that glucose uptake and glycolysis decreased in Nek1^{Kat2J/Kat2J} MEFs. We have also rephrased some conclusions to avoid overinterpretations.

Minor points

-The introduction is too long and masks the workflow of this study.

Reply: We have modified the Introduction and Discussion to make them more succinct.

-some statistics data are often missing in the results part and it could be helpful to provide some of them in the text, thus avoiding to systematically have a look on the figures.

Reply: Will try to add statistics data in the text.

-How could the authors explain the dispersion of the results in figure 3 d within the subgroup of Nek1 Kat2/Kat2J? In the figure 3, some inflammation markers are measured but their specific role is not clear.

Reply:

Figure 3d shows a fraction of neurons in motor cortex showed increased α -Synuclein protein levels in Nek1^{Kat2J/Kat2J} mice brain. Considering the high diversity of neurons in the brain, these data suggest that neurons in motor cortex tend to have more α -synuclein accumulation in Nek1^{Kat2J/Kat2J} mice.

The inflammation markers in figure 3 are transcriptionally regulated by RIPK1 kinase activity in neuroinflammation in previously published papers (Ito et al., 2016; Xu et al., 2018).

-seahorse findings are not explained enough in the context of other metabolic data

Reply: The metabolic ^{13}C -glucose labelling experiment demonstrated the dramatically reduced uptake of glucose in Nek1^{Kat2J/Kat2J} MEFs. As a result, the glycolysis activity and glycolysis intermediate metabolites were also significantly reduced in Nek1^{Kat2J/Kat2J} MEFs. With reduced precursor metabolite such as pyruvate from glycolysis (Extended Data Fig. 5c), the TCA activity of Nek1^{Kat2J/Kat2J} MEFs was also compromised, which were evidenced by the reduced the oxygen consumption rate (Extended Data Fig. 5d) and level of acetyl-coA (Extended Data Fig. 5e) in Nek1^{Kat2J/Kat2J} MEFs.

-Have the authors evaluated the role of pentose phosphate pathway in their experiments to see the proportion between glycolysis and PPP?

Reply: Since our focus on the immediate consequence of glucose uptake defect in Nek1^{Kat2J/Kat2J} MEFs, we did not conduct labeling tracing analysis of PPP, which requires a special labeling product that is difficult for us to obtain.

-What are the main criteria to highlight candidates in the volcano plot (fig 4)?

Reply: P value from multiple t-test was calculated among WT MEFs and Nek1^{Kat2J} MEFs, adjusted $p < 0.05$ was used as the criteria to highlight the hits.

Reviewer #3 (Remarks to the Author):

The manuscript by Wang et al provides critical evidence for NEK1's functions in regulating retromer, blood brain barrier (BBB) integrity, glucose metabolism and RIPK1 activation. NEK1, a serine/threonine kinase, is a risk gene for ALS. This paper shows that NEK1 phosphorylates VPS26B, a component of retromer; and deficiency in NEK1 results in multiple phenotypes, including disrupted endosomal trafficking of plasma membrane proteins, dysfunctional mitochondria and lysosomes, accumulation of α -synuclein, RIPK1 activation, and impaired BBB. Inactivation of RIPK1 or feeding with ketogenic diet in NEK1 deficit mice can prevent their postnatal lethality and BBB damage. These results reveal a novel regulatory mechanism of retromer complex and its function, and uncover insights into NEK1 regulation of ALS pathogenesis. While this paper is of importance to the field, a few concerns remain to be addressed.

Reply: We thank this reviewer for strong support! We worked hard to provide new experimental data to address the concerns raised by this reviewer.

Major concerns:

First, this manuscript covers several proteins' functions in multiple aspects, including NEK1, VPS26b, RIPK1, ... Thus, many gaps in between remain, and many questions raised. For example, it lacks of evidence to functionally link VPS26b with the NEK1's functions in regulating BBB and RIPK1 activation. Does VPS26b deficiency show similar phenotypes as that of NEK1 mutant mice? Does expression of VPS26b or alter VPS26b's phosphorylation diminish the phenotypes in NEK1 mutant mice?

Reply: To begin to address if VPS26b deficiency shows similar phenotypes as that of NEK1 mutant, we provide new experimental data on the VPS26b deficiency and VPS26b's phosphorylation in MEFs as we don't have Vps26b knockout mice available. By addback wt and phosphosite mutants of VPS26b in Vps26b knockdown cells, we evaluated cellular sensitivity to RIPK1-dependent apoptosis and necroptosis, glucose uptake, lysosome size.

Knockdown of VPS26B by shRNA targeting 3' UTR of Vps26b sensitized cells to TNF/5z7 induced RIPK1-dependent apoptosis and TNF/5z7/zVAD induced necroptosis (Fig. 4a-b, Extended Data Fig. 4j-k). While addback of VPS26B^{WT} and VPS26B^{S302D/S304D}, but not VPS26B^{S302A/S304A}, rescued VPS26B knockdown cells from the over-sensitization to these cell death stimuli (Extended Data Fig. 4j-k). Knockdown of VPS26B caused decreased glucose uptake and acetyl-CoA levels in cells. Addback of VPS26B^{WT} and VPS26B^{S302D/S304D} significantly rescued the decreased glucose uptake and acetyl-CoA levels in VPS26B knockdown cells, while addback of VPS26B^{S302A/S304A} could not rescue the defects in glucose uptake and acetyl-CoA levels in VPS26B knockdown cells (Extended Data Fig. 5 i-j; Extended data Fig. 4k). Lysosome sizes increased in VPS26B knockdown cells. Mitochondria fragmentation and increased lysosome sizes in VPS26B knockdown cells were rescued by addback of VPS26B^{WT} and VPS26B^{S302D/S304D}, but not by VPS26B^{S302A/S304A} (Extended Data Fig. 6g-h). In addition to the data in figure 4a showing that knockdown of Vps26b sensitized Nek1^{+/+} but not Nek1^{Kat2J/Kat2J} cells to RIPK1-dependent apoptosis and necroptosis, these data support the concept that VPS26b deficiency show similar phenotypes as that of NEK1^{Kat2J/Kat2J} MEFs. A full characterization of Vps26b knockout mouse phenotype will be done by a future postdoc as this has already been a 5-year+ project.

How does NEK1 regulate RIPK1 activation?

Reply: Our new data shows that the metabolic defects promote RIPK1 activation by reducing the acetylation of A20 in Nek1^{Kat2J/Kat2J} MEFs (in Extended Data Fig. 6i), which in turn promotes lysosomal degradation of A20. Either blocking the lysosomal degradative pathway or promoting fatty acid metabolism by acetyl-L-carnitine and octanoate rescued the decreased A20 protein level in Nek1^{Kat2J/Kat2J} cells (Fig. 2e, 6i; Extended Data Fig. 6j). We have recently reported that reduced acetylation of A20 causes its degradation through the lysosome pathway (Zou et al., 2020). A20 is a key modulator of RIPK1 activation and A20 deficiency in different lineages model different human inflammatory diseases (Mifflin et al., 2020).

Second, this study showed that Nek1Kat2j/Kat2j reduced the phosphorylation of VPS26B (Fig 4e); the non-phosphomimetic VPS26B (S302A/S304A) increased its binding with SNX27 (a retromer complex that promotes surface levels of GLUT1)(...), but decreased its binding with SNX2 (Fig 4g); and the binding of phosphomimetic VPS26B (S302D/S304D) with SNX27 was reduced. However, Nek1Kat2j/Kat2j also showed significantly reduced plasma membrane distribution of GLUT1, while the SNX27 and VPS26B interaction was increased. How to explain these results? It would be better to illustrate clearly the exactly working model or hypothesis.

Reply: To further demonstrate the mechanism by which NEK1 regulates retromer function by phosphorylating VPS26B, we analyzed the effect of VPS26B^{S302A/S304A} and VPS26B^{S302D/S304D} on the trafficking of SNX27 and SNX2. Consistent with increased plasma membrane presence of SNX27 in Nek1^{Kat2J/Kat2J} MEFs compared to that of WT, the levels of SNX27 in the cell surface membrane were increased in Nek1 knockdown MEFs expressing VPS26B^{S302A/S304A} and reduced in cells expressing VPS26B^{S302D/S304D}. In contrast, the levels of SNX2 in the cell surface membrane were decreased in VPS26B knockdown MEFs expressing VPS26B^{S302A/S304A} and increased in VPS26B knockdown MEFs expressing VPS26B^{S302D/S304D} (Extended Data Fig. 4l). In addition, we further confirmed the increased and decreased plasma membrane levels of SNX27 and SNX2, respectively, in Nek1^{Kat2J/Kat2J} MEFs compared to that of WT by western blotting analysis after cell surface selective aminoxy-biotinylation (Fig. 5a). Together with the previous data, these results allow us to propose a model for the molecular mechanism that Nek1 deficiency blocks the phosphorylation of S302/S304 VPS26B which in turn leads to mislocalization of SNX27- and SNX2-retromers (Extended Data Fig. 7g). Phosphorylation of S302/S304 VPS26B by NEK1 modulates the recycling of SNX27-retromer and SNX2-retromer that deliver proteins from endosomes to cell surface membrane and lysosomes. Thus, NEK1 deficiency blocks the recycling of SNX27-retromer from cell surface to endosomes which makes less SNX27-retromer available for delivering substrates such as GLUT1 to cell surface membrane. Since SNX27-retromer is important for delivering GLUT1 to the cell surface membrane, NEK1 deficiency leads to reduced GLUT1 levels at the cell surface membrane which reduces glucose uptake. Reduced glucose uptake disrupts mitochondrial metabolism and reduces cellular levels of acetyl-CoA. Reduced acetyl-CoA inhibits the acetylation of A20 which promotes its lysosomal degradation (Zou et al., 2020). Reduced levels of A20 sensitizes to the activation of RIPK1-dependent cell death and inflammation which can be inhibited by Nec-1s and also by ketogenic diet which provides alternate source of acetyl-CoA to restore cellular levels of A20.

Other concerns:

1. There are two panels ‘e’ in the extended Fig.1, the second one should be ‘f’.

Reply: Fixed.

2. Fig 1b only shows the survival rate of 3 groups of mice, the 3 missing groups should be included.

Reply: The mice in the $Nek1^{Kat2J/+}$, $Nek1^{+/+};RIPK1^{D138N/D138N}$ and $Nek1^{Kat2J/+};RIPK1^{D138N/D138N}$ group showed no premature death, so the survive curve of these 3 groups overlapped in the figure. We redraw this figure to present the missing groups in Fig 1b.

3. The molecular weight of each band should be marked on the western blot images. For example, some proteins in Fig 2 have multiple bands, which is confusing.

Reply: We added molecular weight of related protein markers for all western blot results in the figures.

4. The description of Fig 2i is unclear. The manuscript (Page 9) describes that ‘a reduction of A20 protein in $Nek1^{Kat2j/Kat2j}$ compared to WT mice’, while the figure shows ‘ $Nek1^{+/+};RIPK1^{D138N/D138N}$ ’ compared to ‘ $Nek1^{Kat2j/Kat2j};RIPK1^{D138N/D138N}$ ’.

Reply: This has been corrected

5. Fig 3a only shows a small area of thioflavin S staining, and it is difficult to reflect the change in the number of thioflavin S+ cells. Could the author show a larger brain area to include more positive cells?

Reply: We now present images of larger brain area to include more positive cells of thioflavin S+ staining in the new version of figures.

6. In Fig 3c, compared with $Nek1^{+/+}$ group, $Nek1^{Kat2j/Kat2j};RIPK1^{D138N/D138N}$ showed obviously more a-Synuclein positive cells, however, this appears to be in-consistent with the quantification data in Fig 3d.

Reply: We replaced the misleading images with correct images.

References:

- Andrade, E.B., Magalhaes, A., Puga, A., Costa, M., Bravo, J., Portugal, C.C., Ribeiro, A., Correia-Neves, M., Faustino, A., Firon, A., *et al.* (2018). A mouse model reproducing the pathophysiology of neonatal group B streptococcal infection. *Nat Commun* 9, 3138.
- Chen, A.Q., Fang, Z., Chen, X.L., Yang, S., Zhou, Y.F., Mao, L., Xia, Y.P., Jin, H.J., Li, Y.N., You, M.F., *et al.* (2019a). Microglia-derived TNF- α mediates endothelial necroptosis aggravating blood brain-barrier disruption after ischemic stroke. *Cell Death Dis* 10, 487.
- Chen, G., Zhou, J., Chen, J., Zhu, J., Liu, S.C., Ding, X.F., and Zhang, Q. (2019b). VHL regulates NEK1 via both HIF-2 α pathway and ubiquitin-proteasome pathway in renal cancer cell. *Biochem Biophys Res Commun* 509, 797-802.
- Choi, M., Ku, T., Chong, K., Yoon, J., and Choi, C. (2011). Minimally invasive molecular delivery into the brain using optical modulation of vascular permeability. *Proc Natl Acad Sci U S A* 108, 9256-9261.
- Clairfeuille, T., Mas, C., Chan, A.S., Yang, Z., Tello-Lafoz, M., Chandra, M., Widagdo, J., Kerr, M.C., Paul, B., Merida, I., *et al.* (2016). A molecular code for endosomal recycling of phosphorylated cargos by the SNX27-retromer complex. *Nat Struct Mol Biol* 23, 921-932.
- Daneman, R., Zhou, L., Kebede, A.A., and Barres, B.A. (2010). Pericytes are required for blood-brain barrier integrity during embryogenesis. *Nature* 468, 562-566.
- Fry, A.M., O'Regan, L., Sabir, S.R., and Bayliss, R. (2012). Cell cycle regulation by the NEK family of protein kinases. *J Cell Sci* 125, 4423-4433.
- Haruwaka, K., Ikegami, A., Tachibana, Y., Ohno, N., Konishi, H., Hashimoto, A., Matsumoto, M., Kato, D., Ono, R., Kiyama, H., *et al.* (2019). Dual microglia effects on blood brain barrier permeability induced by systemic inflammation. *Nat Commun* 10, 5816.
- Ito, Y., Ofengeim, D., Najafov, A., Das, S., Saberi, S., Li, Y., Hitomi, J., Zhu, H., Chen, H., Mayo, L., *et al.* (2016). RIPK1 mediates axonal degeneration by promoting inflammation and necroptosis in ALS. *Science* 353, 603-608.
- Jiang, H., Gallet, S., Klemm, P., Scholl, P., Folz-Donahue, K., Altmuller, J., Alber, J., Heilinger, C., Kukat, C., Loyens, A., *et al.* (2020). MCH Neurons Regulate Permeability of the Median Eminence Barrier. *Neuron* 107, 306-319 e309.
- Kenna, K.P., van Doormaal, P.T., Dekker, A.M., Ticozzi, N., Kenna, B.J., Diekstra, F.P., van Rheenen, W., van Eijk, K.R., Jones, A.R., Keagle, P., *et al.* (2016). NEK1 variants confer susceptibility to amyotrophic lateral sclerosis. *Nat Genet* 48, 1037-1042.
- Keren-Shaul, H., Spinrad, A., Weiner, A., Matcovitch-Natan, O., Dvir-Szternfeld, R., Ulland, T.K., David, E., Baruch, K., Lara-Astaiso, D., Toth, B., *et al.* (2017). A Unique Microglia Type Associated with Restricting Development of Alzheimer's Disease. *Cell* 169, 1276-1290 e1217.
- Lin, Y., Pan, Y., Wang, M., Huang, X., Yin, Y., Wang, Y., Jia, F., Xiong, W., Zhang, N., and Jiang, J.Y. (2012). Blood-brain barrier permeability is positively correlated with cerebral microvascular perfusion in the early fluid percussion-injured brain of the rat. *Lab Invest* 92, 1623-1634.
- Mao, L., Liao, C., Qin, J., Gong, Y., Zhou, Y., Li, S., Liu, Z., Deng, H., Deng, W., Sun, Q., *et al.* (2021). Phosphorylation of SNX27 by MAPK11/14 links cellular stress-signaling pathways with endocytic recycling. *J Cell Biol* 220.
- McInerney-Leo, A.M., Harris, J.E., Leo, P.J., Marshall, M.S., Gardiner, B., Kinning, E., Leong, H.Y., McKenzie, F., Ong, W.P., Vodopiutz, J., *et al.* (2015). Whole exome sequencing is an efficient, sensitive and specific method for determining the genetic cause of short-rib thoracic dystrophies. *Clin Genet* 88, 550-557.
- Mifflin, L., Ofengeim, D., and Yuan, J. (2020). Receptor-interacting protein kinase 1 (RIPK1) as a therapeutic target. *Nat Rev Drug Discov* 19, 553-571.
- Minutti, C.M., Modak, R.V., Macdonald, F., Li, F., Smyth, D.J., Dorward, D.A., Blair, N., Husovsky, C., Muir, A., Giampazolias, E., *et al.* (2019). A Macrophage-Pericyte Axis Directs

Tissue Restoration via Amphiregulin-Induced Transforming Growth Factor Beta Activation. *Immunity* 50, 645-654 e646.

Ofengeim, D., Mazzitelli, S., Ito, Y., DeWitt, J.P., Mifflin, L., Zou, C., Das, S., Adiconis, X., Chen, H., Zhu, H., *et al.* (2017). RIPK1 mediates a disease-associated microglial response in Alzheimer's disease. *Proc Natl Acad Sci U S A* 114, E8788-E8797.

Perga, S., Martire, S., Montarolo, F., Navone, N.D., Calvo, A., Fuda, G., Marchet, A., Leotta, D., Chio, A., and Bertolotto, A. (2017). A20 in Multiple Sclerosis and Parkinson's Disease: Clue to a Common Dysregulation of Anti-Inflammatory Pathways? *Neurotoxicity research* 32, 1-7.

Reyahi, A., Nik, A.M., Ghiami, M., Gritli-Linde, A., Pontén, F., Johansson, B.R., and Carlsson, P. (2015). Foxf2 Is Required for Brain Pericyte Differentiation and Development and Maintenance of the Blood-Brain Barrier. *Dev Cell* 34, 19-32.

Singh, V., Connelly, Z.M., Shen, X., and De Benedetti, A. (2017). Identification of the proteome complement of humanTLK1 reveals it binds and phosphorylates NEK1 regulating its activity. *Cell Cycle* 16, 915-926.

Tager, A.M., LaCamera, P., Shea, B.S., Campanella, G.S., Selman, M., Zhao, Z., Polosukhin, V., Wain, J., Karimi-Shah, B.A., Kim, N.D., *et al.* (2008). The lysophosphatidic acid receptor LPA1 links pulmonary fibrosis to lung injury by mediating fibroblast recruitment and vascular leak. *Nat Med* 14, 45-54.

Thiel, C., Kessler, K., Giessler, A., Dimmler, A., Shalev, S.A., von der Haar, S., Zenker, M., Zahnleiter, D., Stoss, H., Beinder, E., *et al.* (2011). NEK1 mutations cause short-rib polydactyly syndrome type majewski. *Am J Hum Genet* 88, 106-114.

Xu, D., Jin, T., Zhu, H., Chen, H., Ofengeim, D., Zou, C., Mifflin, L., Pan, L., Amin, P., Li, W., *et al.* (2018). TBK1 Suppresses RIPK1-Driven Apoptosis and Inflammation during Development and in Aging. *Cell* 174, 1477-1491 e1419.

Zou, C., Mifflin, L., Hu, Z., Zhang, T., Shan, B., Wang, H., Xing, X., Zhu, H., Adiconis, X., Levin, J.Z., *et al.* (2020). Reduction of mNAT1/hNAT2 Contributes to Cerebral Endothelial Necroptosis and Abeta Accumulation in Alzheimer's Disease. *Cell reports* 33, 108447.

REVIEWERS' COMMENTS

Reviewer #1 (Remarks to the Author):

The authors adequately addressed most of my concerns. Hence, I agree with publication of this study in NCOMM.

Reviewer #2 (Remarks to the Author):

I thank the authors for all the comments they have provided and for the several experiments they have added to answer my questions. The quantity of data is impressive and appropriate. My only request is to moderate the scope of the results toward ALS perspectives.

Reviewer #3 (Remarks to the Author):

The revised manuscript has addressed my concerns. No more issues.